# A novel role of DNA polymerase λ in translesion synthesis in conjunction with DNA polymerase ζ

Jung-Hoon Yoon, Debashree Basu, Karthi Sellamuthu, Robert E Johnson, Satya Prakash, Louise Prakash

By extending synthesis opposite from a diverse array of DNA lesions, DNA polymerase (Pol) ζ performs a crucial role in translesion synthesis (TLS). In yeast and cancer cells, Rev1 functions as an indispensable scaffolding component of Polζ and it imposes highly error-prone TLS upon Polζ. However, for TLS that occurs during replication in normal human cells, Rev1 functions instead as a scaffolding component of Pols η, ι, and κ and Rev1-dependent TLS by these Pols operates in a predominantly error-free manner. The lack of Rev1 requirement for Polζ function in TLS in normal cells suggested that some other protein substitutes for this Rev1 role. Here, we identify a novel role of Polλ as an indispensable scaffolding component of Polζ. TLS studies opposite a number of DNA lesions support the conclusion that as an integral component, Polλ adapts Polζ-dependent TLS to operate in a predominantly error-free manner in human cells, essential for genome integrity and cellular homeostasis.

## Introduction

DNA lesions that escape removal by excision repair processes block the progression of replication forks (RFs). By promoting proficient replication through DNA lesions, translesion synthesis (TLS) DNA polymerases (Pols) prevent the collapse of RFs stalled at DNA lesion sites and protect against chromosomal instability and tumorigenesis (Yoon et al, 2019b).

Biochemical, structural, and genetic studies have indicated that TLS Pols have specialized functions in replicating through DNA lesions, and that their active sites are adapted for accommodating specific types of DNA lesions (Prakash et al, 2005). For example, because of its unique ability to accommodate two template residues in its active site, Polη can proficiently replicate through the UV-induced covalently linked cyclobutane pyrimidine dimer (CPD) by forming a Watson-Crick (W-C) base pair between each pyrimidine of the CPD and an incoming nucleotide (nt) (Johnson et al, 1999, 2000b; Masutani et al, 1999; Washington et al, 2000, 2003; Biertumpfel et al, 2010; Silverstein et al, 2010). Whereas replication through certain DNA

lesions can be performed by the action of just one Pol, such as by Polη opposite CPDs, replication through a vast array of other DNA lesions requires the sequential action of two TLS Pols, wherein one Pol inserts a nt opposite the DNA lesion site from which another Pol extends synthesis (Johnson et al, 2000a; Prakash & Prakash, 2002; Prakash et al, 2005). Biochemical studies have shown that Polζ functions specifically in the extension step of TLS and is proficient in extending synthesis past a large variety of DNA lesions (Johnson et al, 2000a, 2001; Haracska et al, 2001; Nair et al, 2006).

In yeast, Rev1 functions as an indispensable scaffolding component of Polζ, and its ability to associate—via its C terminus—with Polζ is essential for Polζ's ability to function in TLS in yeast cells, and TLS by Rev1/Polζ complex operates in a highly mutagenic manner (Baynton et al, 1999; Nelson et al, 2000; Haracska et al, 2001; Gibbs et al, 2005; Acharya et al, 2006). In normal human cells, however, Rev1 does not function together with Polζ; instead, Rev1 functions as an indispensable scaffolding component of the Y-family DNA Pols η, ι, and κ (Yoon et al, 2015). Moreover, Rev1-dependent TLS by Y-family Pols operates in a much more error-free manner in human cells than would be expected from the low fidelity of the purified Pols. We have suggested previously that as a scaffolding component, Rev1 effects the assembly of a multiprotein complex wherein the fidelity of the TLS Polη, ι, κ, or Rev1 is enhanced (Yoon et al, 2019a, 2018, 2017, 2015).

The lack of Rev1 requirement for Polζ function in TLS that operates during replication in normal human cells (Yoon et al, 2015) suggested that some other protein substitutes for Rev1 in this role. Here we identify a role for Polλ, an X-family DNA Pol, in Polζ-dependent TLS in normal human cells. Previous studies have implicated Polλ in short patch base excision repair and non-homologous end joining (Garcia-Diaz et al, 2001, 2005; Lee et al, 2004; Braithwaite et al, 2005a, 2005b, 2010; Pryor et al, 2015) and purified Polλ can catalyze synthesis across an abasic site and a thymine glycol (Maga et al, 2002; Belousova et al, 2010). Our evidence that Polλ functions as an essential scaffolding component of Polζ adds a novel dimension to Polζ-mediated TLS in normal human cells. Importantly, in contrast to the highly error-prone Rev1/Polζ–mediated TLS that is observed in yeast (Lawrence & Christensen, 1978, 1979; Lawrence et al, 1984; Gibbs et al, 2005; Acharya et al, 2006) or in TLS that occurs

Department of Biochemistry and Molecular Biology, University of Texas Medical Branch, Galveston, TX, USA

Correspondence: loprakas@utmb.edu
Debashree Basu's present address is Ultragenyx Pharmaceuticals, Inc., Woburn, MA, USA

in mammalian cells during gap repair (Yoon et al, 2015), or in cancer cells (Doles et al, 2010; Xie et al, 2010; Xu et al, 2013), TLS dependent upon Polλ/Polζ operates in a predominantly error-free manner in human cells. We discuss the implications of adoption of Polλ as an integral component of Polζ for error-free TLS mediated by the Polλ/Polζ complex.

## Results

### Requirement of Polλ for TLS opposite UV lesions

To analyze the role of Polλ in TLS, we used the SV40 duplex plasmid in which bidirectional replication initiates from an origin of replication and TLS through the DNA lesion carried on the template for leading or lagging strand replication is determined by the frequency of blue colonies among the total *Kan⁺* colonies (Yoon et al, 2009, 2010b). As determined in previous studies, the genetic control of TLS determined by using this plasmid system reflects the TLS mechanisms that operate during genomic replication. For example, using the EBV origin–based plasmid, where plasmid replication is controlled by the same processes that govern cellular replication, we showed that the same TLS Pols are required for replication through UV lesions in the EBV-based plasmid as in the SV40-based plasmid (Yoon et al, 2012). In other studies we have provided evidence that the Pols identified to have a role in TLS opposite UV lesions in plasmid studies are also required for RF progression through UV lesions in primary cells (Yoon et al, 2019b). Although not all the complexities in the genomic context such as topology, chromatin state, or epigenetic modifications will be reflected in the plasmid system, the basic TLS mechanisms remain the same in the genomic context as in the plasmid system (Yoon et al, 2019b).

### TLS opposite cis-syn TT dimer

In our previous analyses of the genetic control of TLS opposite UV lesions, we have shown that TLS through a *cis-syn* TT dimer operates via a Polη-dependent error-free pathway or via an alternative Polθ-dependent pathway which acts in an error-prone manner (Yoon et al, 2019b). Because of its proficient ability to insert nts opposite both the 3′T and 5′T of the dimer and to subsequently extend synthesis, Polη alone conducts TLS through the *cis-syn* TT dimer. In the Polθ-dependent pathway, however, following nt insertion opposite the 3′T of the dimer by Polθ, either Polκ or Polζ is required to insert a nt opposite the 5′T and extend subsequent synthesis (Yoon et al, 2019b) (see also Fig 1A). A role of Polλ in TLS in conjunction with Polζ implies that an epistatic relationship would exist between them.

As shown in Table 1, TLS opposite a *cis-syn* TT dimer carried on the leading strand template in nucleotide excision repair defective XPA cells occurs with a frequency of ~38%. TLS frequency is reduced to ~18% and ~21% in cells depleted for Polη or Polθ, respectively. Depletion of Polκ or the Rev3 or Rev7 subunit of Polζ reduces TLS frequency to ~30% and a similar reduction in TLS frequency occurs in cells depleted for Polλ. Our results that co-depletion of Polκ with Polλ reduces TLS frequency to ~18% conform with a role for Polκ and

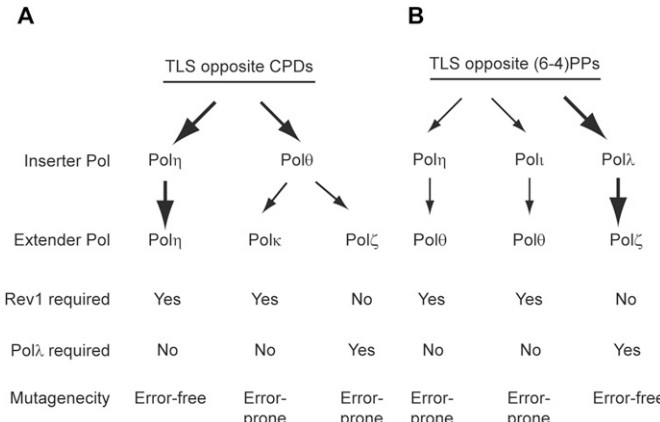

**Figure 1. TLS pathways for replication through UV lesions.**
**(A)** TLS pathways for replication through cyclobutane pyrimidine dimers (CPDs). Polη replicates through CPDs in an error-free manner. Polθ functions in inserting nts opposite the 3′T or 3′C of a CPD in the error-prone TLS pathways dependent upon Polκ or Polζ for extension of synthesis. As a scaffolding component, Rev1 functions together with Polη or Polκ, and Polλ functions together with Polζ. **(B)** TLS pathways for replication through (6-4)PPs. Polθ extends synthesis from the nts inserted by Polη or Polι opposite the 3′T or 3′C of the (6-4) photoproduct and these pathways promote error-prone TLS. After nt insertion opposite the (6-4) lesion by Polλ, Polζ would extend synthesis from the nt inserted by Polλ. The Polλ/Polζ pathway conducts error-free TLS through (6-4) PPs. The thickness of the arrows depicts the relative contribution of TLS Pols to lesion bypass.

Polλ in alternative TLS pathways, and the observation that TLS frequency remains the same in cells co-depleted for Rev3 with Polλ as in cells depleted for either Pol alone, supports a role for Polλ in the same TLS pathway as Polζ. Furthermore, the observation that TLS frequency remains the same in cells co-depleted for Polθ with Polλ (~19%) as in cells depleted for Polθ alone is congruent with epistasis of Polθ over Polλ, similar to the epistasis of Polθ over Polζ we reported previously (Yoon et al, 2019b). In addition, the reduction in TLS frequency to ~12% in cells co-depleted for Polη and Polλ concurs with their role in alternative TLS pathways. Because of the requirement of Rev1 for Polη- and Polκ-dependent TLS pathways, TLS by both these pathways is inhibited in Rev1-depleted cells, and only the Polζ pathway remains functional (Yoon et al, 2015). Our results that TLS frequency is reduced to a residual level of ~4% in cells co-depleted for Rev1 and Polλ (Table 1), similar to those seen upon co-depletion of Rev1 with Polζ (Yoon et al, 2015), support a role of Polλ in Polζ-dependent TLS. We also carried out studies to analyze Polλ's role in TLS opposite a *cis-syn* TT dimer carried on the lagging strand template in the duplex plasmid (Table 1). The epistatic effects of Rev3 and Polλ depletion on TLS frequency and a drastic reduction in TLS frequency in cells co-depleted for Rev1 and Polλ add further evidence for a role of Polλ in TLS in conjunction with Polζ in a pathway that functions independently of Rev1-dependent TLS pathways (Table 1 and Fig 1A).

In addition, we analyzed Polλ's role in TLS opposite a *cis-syn* TT dimer in XPV HFs defective in Polη, and in Polλ⁻/⁻ MEFs (Table 2). In XPV cells, TLS opposite a *cis-syn* TT dimer occurs with a frequency of ~11% and TLS frequency is reduced to ~5% in cells depleted for Polζ (Rev3 or Rev7), Polλ, or co-depleted for both Pols (Table 2). In Polλ⁻/⁻ MEFs, TLS opposite a *cis-syn* TT dimer occurs at a frequency of ~24%,

**Table 1.   The effects of siRNA knockdowns of Polλ and other TLS Pols on replicative bypass of a *cis-syn* TT dimer or a (6-4) TT photoproduct carried on the leading or lagging DNA strand template in XPA human fibroblasts.**

| UV lesion | siRNA | Leading strand | | | Lagging strand | | |
|---|---|---|---|---|---|---|---|
| | | Number of *Kan⁺* colonies | Number of blue colonies among *Kan⁺* | TLS (%)[a] | Number of *Kan⁺* colonies | Number of blue colonies among *Kan⁺* | TLS (%)[a] |
| *cis-syn* TT dimer | NC | 534 | 205 | 38.4 | 465 | 152 | 32.7 |
| | Polη | 406 | 72 | 17.7 | 389 | 62 | 15.9 |
| | Polθ | 412 | 86 | 20.9 | 394 | 60 | 15.2 |
| | Polκ | 528 | 158 | 29.9 | 412 | 89 | 21.6 |
| | Rev3 | 436 | 127 | 29.1 | 408 | 81 | 19.9 |
| | Rev7 | 512 | 146 | 28.5 | 570 | 124 | 21.8 |
| | Polλ | 508 | 144 | 28.3 | 394 | 90 | 22.8 |
| | Rev1 | 446 | 66 | 14.8 | 366 | 45 | 12.3 |
| | Polκ + Polλ | 316 | 56 | 17.7 | 543 | 86 | 15.8 |
| | Rev3 + Polλ | 566 | 162 | 28.6 | 612 | 124 | 20.3 |
| | Polθ + Polλ | 306 | 59 | 19.3 | 405 | 65 | 16.0 |
| | Polη + Polλ | 562 | 66 | 11.7 | 587 | 56 | 9.5 |
| | Rev1 + Polλ | 408 | 18 | 4.4 | 422 | 15 | 3.6 |
| (6-4) TT PP | NC | 630 | 215 | 34.1 | 562 | 174 | 31.0 |
| | Polλ | 386 | 65 | 16.8 | 482 | 71 | 14.7 |
| | Rev3 | 426 | 73 | 17.1 | 685 | 100 | 14.6 |
| | Rev7 | 506 | 82 | 16.2 | 541 | 84 | 15.5 |
| | Rev3 + Polλ | 401 | 67 | 16.7 | 612 | 94 | 15.4 |
| | Rev1 | 402 | 64 | 15.9 | 366 | 68 | 18.6 |
| | Polθ | 468 | 84 | 17.9 | 556 | 88 | 15.8 |
| | Rev1 + Polλ | 545 | 28 | 5.1 | 504 | 24 | 4.8 |
| | Polθ + Polλ | 668 | 37 | 5.5 | 635 | 30 | 4.7 |

[a]Statistical analyses of these data are shown in Table S1.

and TLS frequency remains the same in Rev3 depleted Polλ⁻/⁻ MEFs, whereas Polη depletion reduces TLS frequency to ~11%. (Table 2). The epistatic effects of Polλ and Polζ depletion in XPV HFs and of Polζ depletion in Polλ⁻/⁻ MEFs add further evidence for a role of Polλ in TLS in conjunction with Polζ.

### TLS opposite (6-4) TT photoproduct

In previous studies, we provided evidence that TLS opposite a (6-4) TT photoproduct occurs via two error-prone pathways dependent upon either Polη/Polθ or Polι/polθ in which after nt insertion opposite the 3'T of the photoproduct by Polη or Polι, Polθ would insert a nt opposite the 5'T and extend synthesis further (Yoon et al, 2019b) (Fig 1B). As a scaffolding component, Rev1 is required for TLS by both Polη and Polι. Polζ functions in an alternative pathway independent of Polθ and Rev1, and it promotes error-free TLS through (6-4) PPs (Yoon et al, 2019b, 2015) (Fig 1B). Because purified Polζ lacks the ability to insert a nt opposite the 3'T of (6-4) TT but extends synthesis from the nt inserted opposite the 3'T of the photoproduct (Johnson et al, 2000a, 2001), we have previously suggested that another TLS

Pol, which had remained to be identified, would insert a nt opposite the 3'T of (6-4) TT from which Polζ would extend synthesis (Yoon et al, 2010b). In the sections below, we provide evidence for such a role of Polλ.

As shown in Table 1, in XPA HFs, TLS opposite a (6-4) TT PP carried on the leading strand template occurs with a frequency of ~34%, and TLS frequency declines to ~17% in cells depleted either for Polλ, Rev3, or Rev7, or co-depleted for Rev3 and Polλ The epistatic relationship of Polλ with Polζ implies that Polλ functions together with Polζ in TLS at (6-4) PPs. In cells depleted for Rev1 or Polθ, TLS opposite (6-4) TT PP occurs at a frequency of ~16–18% and co-depletion of Polλ with either Rev1 or Polθ confers a drastic reduction in TLS to a level of ~5% (Table 1), similar to the reduction in TLS that occurs upon the depletion of Polζ with Rev1 or with Polθ (Yoon et al, 2019b, 2015). These data are congruent with a role of Polλ in conjunction with Polζ in a pathway that operates independently of TLS dependent upon Rev1 and Polθ. We additionally verified these conclusions for TLS opposite a (6-4) TT PP carried on the lagging strand template in XPA HFs (Table 1). Also, the epistatic relationship between Polλ and Polζ observed in XPV HFs or Polλ⁻/⁻ MEFs (Table 2) adds further evidence for a role of Polλ in promoting

**Table 2. Effects of siRNA knockdowns of Polλ and other TLS Pols on replicative bypass of a *cis-syn* TT dimer or a (6-4) TT photoproduct carried on the leading DNA strand template in XPV HFs or in WT or Polλ⁻/⁻ MEFs.**

| Cells | UV lesion | siRNA | Number of *Kan*⁺ colonies | Number of blue colonies among *Kan*⁺ | TLS (%)[a] |
|---|---|---|---|---|---|
| XPV HFs | *cis-syn* TT dimer | NC | 312 | 34 | 10.9 |
| | | Rev3 | 308 | 17 | 5.5 |
| | | Rev7 | 321 | 18 | 5.6 |
| | | Polλ | 274 | 16 | 5.8 |
| | | Rev3 + Polλ | 332 | 19 | 5.7 |
| | | Rev7 + Polλ | 294 | 16 | 5.4 |
| XPV HFs | (6-4) TT PP | NC | 405 | 62 | 15.3 |
| | | Polθ | 294 | 23 | 7.8 |
| | | Rev3 | 348 | 25 | 7.2 |
| | | Polλ | 355 | 26 | 7.3 |
| | | Rev3 + Polλ | 304 | 25 | 8.2 |
| | | Polθ + Polλ | 278 | 10 | 3.6 |
| WT MEFs | *cis-syn* TT dimer | NC | 376 | 120 | 31.9 |
| Polλ⁻/⁻ MEFs | | NC | 432 | 104 | 24.1 |
| | | Rev3 | 295 | 70 | 23.7 |
| | | Polη | 248 | 27 | 10.9 |
| WT MEFs | (6-4) TT PP | NC | 422 | 128 | 30.3 |
| Polλ⁻/⁻ MEFs | | NC | 390 | 57 | 14.6 |
| | | Rev3 | 276 | 42 | 15.2 |
| | | Rev1 | 388 | 10 | 2.6 |
| | | Polθ | 408 | 11 | 2.7 |

[a]Statistical analyses of these data are shown in Table S2.

TLS through (6-4) TT photoproduct via a Polζ-dependent pathway that operates independently of Rev1 and Polθ (Fig 1B).

### Non-catalytic and catalytic roles of Polλ for TLS opposite UV lesions

A scaffolding role of Polλ in Polζ-dependent TLS implies that for TLS opposite CPDs where Polζ would extend synthesis from the nt inserted opposite the 3′T or 3′C by Polθ, its DNA polymerase activity will not be required. For TLS opposite (6-4) PPs, however, Polλ's DNA polymerase activity could be required for nt insertion opposite the 3′T or 3′C of the photoproduct because the polymerase for such a role has not been identified previously (Yoon et al, 2010b, 2019b). To establish that Polλ polymerase activity was not required for TLS opposite CPDs and to determine if its polymerase function was required for TLS opposite (6-4) PP, we analyzed the effects of the D427A, D429A mutations which inactivate Polλ's DNA polymerase activity. As shown in Table 3, TLS opposite a *cis-syn* TT dimer in Polλ⁻/⁻ MEFs carrying the vector plasmid occurs with a frequency of ~24% and TLS frequency rises to ~32% in Polλ⁻/⁻ MEFs expressing WT Polλ (Fig S1B). Our results that TLS frequency remains the same in Polλ⁻/⁻ MEFs, which express the Polλ catalytic mutant protein (Fig S1B) as in cells expressing WT Polλ (Table 3) validate the conclusion that Polλ's polymerase activity is not required for TLS opposite a *cis-*

*syn* TT dimer. Thus, for TLS opposite CPDs, only the non-catalytic scaffolding role of Polλ is required.

TLS opposite a (6-4) TT photoproduct in Polλ⁻/⁻ MEFs carrying the vector plasmid occurs with a frequency of ~15% and TLS frequency rises to ~30% in Polλ⁻/⁻ MEFs expressing WT Polλ (Table 3). Our observation that TLS frequency remains at ~14% in Polλ⁻/⁻ MEFs which express the Polλ catalytic mutant provides strong evidence for the requirement of Polλ DNA polymerase activity for TLS opposite (6-4) photoproduct (Table 3). Thus, for TLS opposite (6-4) PPs, both the DNA synthetic and the non-catalytic scaffolding roles of Polλ would be required.

### Biochemical analysis of Polλ's role in TLS opposite (6-4) TT photoproduct

Because the DNA synthesis activity of Polλ is required for TLS opposite the (6-4) photoproduct in human cells, we examined whether purified Polλ (Fig 2A) has an ability to insert dATP, dTTP, dGTP, or dCTP opposite the 3′T of a (6-4) TT photoproduct. As shown in Fig 2B, on undamaged DNA, Polλ inserts dATP opposite both the 3′T and 5′T efficiently such that almost all the DNA substrate has been used up, and in the presence of 4 dNTPs, it extends synthesis further. Remarkably, we find that Polλ inserts dATP opposite the 3′T as well as the 5′T of the (6-4) photoproduct, albeit less efficiently. In

**Table 3. Effects of catalytically active (WT) Polλ or catalytically inactive D427A and D429A mutant Polλ on TLS opposite a *cis-syn* TT dimer, a (6-4) TT photoproduct, a thymine glycol, or a 1,N⁶-ethenodeoxyadenosine, carried on the leading strand template in Polλ⁻/⁻ MEFs.**

| DNA lesion | Vector expressing | Number of *Kan*⁺ colonies | Number of blue colonies among *Kan*⁺ colonies | TLS (%)[a] |
|---|---|---|---|---|
| *cis-syn* TT dimer | No Polλ (control) | 355 | 87 | 24.5 |
|  | WT Polλ | 348 | 111 | 31.9 |
|  | Mutant Polλ | 409 | 124 | 30.3 |
| (6-4) TT PP | No Polλ (control) | 308 | 46 | 14.9 |
|  | WT Polλ | 316 | 95 | 30.1 |
|  | Mutant Polλ | 286 | 40 | 14.0 |
| Tg | No Polλ (control) | 446 | 46 | 10.3 |
|  | WT Polλ | 395 | 84 | 21.3 |
|  | Mutant Polλ | 361 | 80 | 22.2 |
| εdA | No Polλ (control) | 311 | 34 | 10.9 |
|  | WT Polλ | 405 | 82 | 20.2 |
|  | Mutant Polλ | 286 | 62 | 21.7 |

[a]Statistical analyses of these data are shown in Table S3.

the presence of all 4 dNTPs, Polλ inserts a nt opposite the 3′T and 5′T of the photoproduct and then inserts a nt opposite the next nucleotide as well (Fig 2B). Thus, Polλ performs synthesis opposite (6-4) TT photoproduct, and does so correctly by inserting an A. As expected from the lack of requirement of Polλ DNA synthesis activity for TLS opposite a *cis-syn* TT dimer in vivo (Table 3), purified Polλ shows no TLS activity opposite the dimer (Fig S2).

### Evidence for Polλ's involvement in error-prone TLS opposite CPDs and in error-free TLS opposite (6-4) photoproducts

In normal cells, TLS opposite a *cis-syn* TT dimer generates ~2% mutant TLS products which harbor a change at the 3′T or 5′T (Table 4). The requirement of Polλ in Polζ-dependent TLS opposite CPDs predicted that similar to Polζ (Yoon et al, 2010b), Polλ would also be required for mutagenic TLS opposite CPDs. We find that opposite a *cis-syn* TT dimer carried on the leading or lagging strand template, the incidence of mutagenic TLS is reduced to 0.7% in Polλ depleted cells (Table 4), similar to the reduction that occurs upon Polζ depletion (Yoon et al, 2010b). This result is congruent with a role of Polλ in Polζ-dependent error-prone TLS opposite a *cis-syn* TT dimer.

TLS opposite a (6-4) TT photoproduct also generates ~2% mutational TLS products which harbor a change at the 3′T or 5′T of the photoproduct. The requirement of Polλ for Polζ-dependent error-free TLS opposite (6-4) PPs predicted that the frequency of mutational TLS products would rise in Polλ-depleted cells; accordingly, we find that the frequency of mutant TLS products increases to ~4–5% in Polλ-depleted cells (Table 4), similar to that seen upon Polζ depletion (Yoon et al, 2010b).

Next, we extended mutational analyses to the *cII* gene integrated into the genome of big blue mouse embryonic fibroblasts (BBMEFs) where we could analyze the effect of Polλ depletion on UV-induced mutations resulting from TLS opposite CPDs and (6-4) PPs formed at the various TT, TC, and CC dipyrimidine sites. The spectrum of mutations induced by UV and other DNA damaging agents in the *cII* gene resembles that determined from sequence analyses of

endogenous chromosomal genes or from whole-genome sequence analyses (You et al, 2001; You & Pfeifer, 2001; Besaratinia & Pfeifer, 2006; Alexandrov et al, 2013; Martincorena et al, 2015). To examine UV-induced mutations resulting from TLS opposite CPDs, the (6-4) PPs are selectively removed from the genome by expressing a (6-4) PP photolyase gene in the BBMEF cell line (You et al, 2001). In this cell line, the frequency of spontaneous mutations in the *cII* gene is ~16 × 10⁻⁵, and this frequency remains the same in cells depleted for Polλ (Table 5). In UV-irradiated (5 J/m²) cells exposed to photo-reactivating light to activate (6-4) PP removal, mutation frequency rises to ~46 × 10⁻⁵ (Table 5). Thus, the additional increase of ~30 × 10⁻⁵ in mutation frequency results from mutagenic TLS opposite

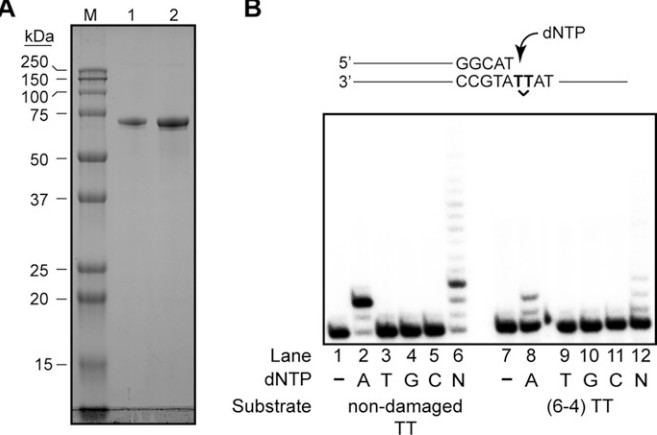

**Figure 2. DNA synthesis opposite (6-4) TT photoproduct by purified Polλ.**
**(A)** SDS–PAGE of purified full-length Polλ. Human DNA polymerase λ was purified from yeast cells as described in the Materials and Methods section. Lane M, 5 μl precision plus protein standard (Bio-Rad). Lane 1, 0.75 μg Polλ. Lane 2, 1.5 μg Polλ. Molecular weight marker sizes are given on the left. **(B)** Nucleotide incorporation opposite an undamaged TT or opposite 6-4 (TT) photoproduct. Reactions were carried out as described in the Materials and Methods section and included either a single deoxynucleotide triphosphate (dGTP, dATP, dTTP, or dCTP) or all 4 dNTPs. A schematic representation of the substrate is shown on top.

**Table 4.** Effects of siRNA knockdown of Polλ on mutation frequencies and nucleotides inserted opposite a *cis-syn* TT dimer or a (6-4) TT photoproduct carried on the leading or lagging strand DNA template in nucleotide excision repair–defective XPA human fibroblasts.

| DNA lesion | Lesion containing DNA strand | siRNA | # of *Kan*$^+$ blue colonies sequenced | A | G | C | T | Other[a] | Mutation frequency % |
|---|---|---|---|---|---|---|---|---|---|
| *cis-syn* TT dimer | Leading | NC | 334 (7)[b] | 327 | 4 (3′ T)[c] | — | 2 (5′ T) / 1 (3′ T) | — | 2.1 |
| | | Polλ | 402 (3) | 399 | 3 (3′ T) | — | — | — | 0.7 |
| | Lagging | NC | 375 (10) | 365 | 4 (5′ T) / 2 (3′ T) | — | 1 (5′ T) / 3 (3′ T) | — | 2.7 |
| | | Polλ | 318 (2) | 316 | 2 (3′ T) | — | — | — | 0.6 |
| (6-4) TT photoproduct | Leading | NC | 342 (8) | 334 | 3 (5′ T) / 2 (3′ T) | — | 2 (3′ T) | 1 | 2.3 |
| | | Polλ | 347 (15) | 332 | 1 (5′ T) / 2 (3′ T) | 6 (3′ T) | 4 (3′ T) | 2 | 4.3 |
| | Lagging | NC | 384 (7) | 377 | 2 (5′ T) / 2 (3′ T) | — | 2 (3′ T) | 1 | 1.8 |
| | | Polλ | 320 (16) | 304 | 4 (5′ T) / 3 (3′ T) | 4 (3′ T) | 3 (3′ T) | 2 | 5.0 |

[a]Mutations at the flanking site.
[b]Numbers of colonies where TLS occurred by insertion of a nucleotide other than an A are shown in parenthesis.
[c]The site where mutation occurred, 3′T or 5′T of the UV lesion is indicated in parenthesis.

CPDs formed at the dipyrimidine sites in the *cII* gene. The highly elevated UV-induced mutation frequency (~90 × 10$^{-5}$) in Polη-depleted cells and the greatly reduced mutation frequency (~17 × 10$^{-5}$) in Polθ-depleted cells results from the requirement of Polη for error-free TLS and from the indispensability of Polθ for error-prone TLS opposite CPDs, respectively (Table 5). Mutation frequency in UV-irradiated cells depleted for Polλ declines to the same level as in Rev3-depleted cells and remains the same in cells co-depleted for Polλ and Rev3 (~32 × 10$^{-5}$), indicating epistasis (Table 5). In addition, the evidence that co-depletion of Rev1 and Polλ reduces mutation frequency in UV-irradiated cells to a level near to that in unirradiated cells is consistent with the requirement of Polλ for Polζ-dependent error-prone TLS opposite CPDs that operates independently of Rev1-dependent TLS (Table 5 and Fig 1A).

To examine UV mutations resulting from TLS opposite (6-4) PPs, CPDs are selectively removed from the genome by expressing a CPD photolyase gene in the BBMEF cell line (You et al, 2001). In this cell line, spontaneous mutations in the *cII* gene occur at a frequency of ~18 × 10$^{-5}$ and Polλ depletion has no perceptible effect on this frequency (Table 5). In UV-irradiated (5 J/m²) cells exposed to photoreactivating light to activate the CPD photolyase, mutation frequency increases to ~29 × 10$^{-5}$, and in cells depleted for Rev1 or Polθ, mutation frequency declines to the same level (~18 × 10$^{-5}$) as in unirradiated cells (Table 5), consistent with the requirement of Rev1 and Polθ for Polη- and Polι-dependent error-prone TLS opposite (6-4) PPs (Fig 1B). As expected from the requirement of Polλ for Polζ-dependent error-free TLS, UV-induced mutation frequency rises to ~40 × 10$^{-5}$ in cells depleted for Polλ or Rev3, or co-depleted for both Pols (Table 5). The epistatic effects of Polλ and Rev3

depletions on error-free TLS opposite (6-4) PPs as well as the observation that UV-induced mutation frequency declines to near spontaneous levels in cells co-depleted for Polλ together with Rev1 or with Polθ (Table 5) provide confirmatory evidence for the involvement of Polλ in Polζ-dependent error-free TLS opposite (6-4) PPs in a pathway that provides an alternative to the error-prone pathway in which Polη or Polι function together with Rev1 and require Polθ for the extension step of TLS (Fig 1B).

In BBMEFs expressing no photolyase, UV-induced mutations resulting from TLS opposite both CPDs and (6-4) PPs would accumulate in the *cII* gene. In this cell line, spontaneous mutations occur at a frequency of ~15 × 10$^{-5}$ and mutation frequency increases to ~55 × 10$^{-5}$ upon UV treatment (5 J/m²) (Table 5). In UV irradiated cells, Polλ depletion reduces mutation frequency to the same level (~35 × 10$^{-5}$) as occurs upon Rev3 depletion, consistent with their contribution to error-prone TLS opposite CPDs which account for a large majority of UV-induced mutations (~80%) (Pfeifer, 1997; Yoon et al, 2000; You et al, 2001).

UV-induced C>T and CC>TT signature mutations resulting from mutagenic TLS through CPDs accumulate in the *cII* gene at mutational hot spots located at 11 dipyrimidine sequences, #1 to #11 in WT cells (Yoon et al, 2009). In Rev3 depleted BBMEFs, mutational hot spots at positions #1, #2, #3, and #6 remain, whereas the mutational hot spots at other positions are abrogated (Fig 3A). Previously, we provided evidence that in their role in extending synthesis from the nt inserted opposite the 3′ pyrimidine of the CPDs by another Pol, Polκ generates a different spectrum of hot spot mutations than Polζ (Yoon et al, 2009); hence the spectrum of hot spot mutations in Polζ depleted cells results from Polκ's role in TLS and vice versa. Our observation that the spectrum of mutational hot spots generated by Polλ depletion is nearly identical to the pattern in Rev3

**Table 5. UV-induced mutation frequencies in the *cII* gene in BBMEF cells expressing a (6-4) PP photolyase, cyclobutane pyrimidine dimer (CPD) photolyase, or no photolyase and treated with siRNA for Polλ or other TLS Pols.**

| Photolyase | siRNA | UV[a] | Photoreactivation[b] | Mutation frequency[c] (×10⁻⁵) |
|---|---|---|---|---|
| (6-4) PP photolyase | NC | – | + | 16.1 ± 0.7 |
| | Polλ | – | + | 16.8 ± 0.7 |
| | NC | + | + | 45.9 ± 1.9 |
| | Polη | + | + | 88.7 ± 1.7 |
| | Polθ | + | + | 17.5 ± 1.1 |
| | Polλ | + | + | 30.1 ± 1.0 |
| | Rev3 | + | + | 31.5 ± 1.6 |
| | Rev1 | + | + | 58.2 ± 2.7 |
| | Rev3 + Polλ | + | + | 31.8 ± 1.1 |
| | Rev1 + Polλ | + | + | 20.1 ± 1.5 |
| CPD photolyase | NC | – | + | 17.9 ± 1.2 |
| | Polλ | – | + | 19.1 ± 1.1 |
| | NC | + | + | 28.9 ± 1.0 |
| | Rev1 | + | + | 18.4 ± 1.6 |
| | Polθ | + | + | 18.4 ± 1.2 |
| | Polλ | + | + | 40.9 ± 1.2 |
| | Rev3 | + | + | 39.1 ± 2.4 |
| | Rev3 + Polλ | + | + | 39.6 ± 1.2 |
| | Rev1 + Polλ | + | + | 20.4 ± 1.6 |
| | Polθ + Polλ | + | + | 21.8 ± 1.4 |
| pNeo vector (no photolyase) | NC | – | + | 14.8 ± 1.0 |
| | Polλ | – | + | 15.0 ± 1.3 |
| | NC | + | + | 55.5 ± 1.9 |
| | Polλ | + | + | 35.0 ± 1.0 |
| | Rev3 | + | + | 36.8 ± 2.9 |

[a]5 J/m² of UVC (254 nm) light.
[b]Photoreactivation with UVA (360 nm) light for 3 h.
[c]Data are represented as mean ± SEM. Mean mutation frequencies and standard error of the mean were calculated from four independent experiments.

depleted cells (Fig 3A) adds further evidence that for TLS opposite CPDs, Polλ functions together with Polζ in the error-prone pathway that operates independently of the Polκ-dependent error-prone pathway.

Mutagenic TLS through (6-4) PPs generates mutational hot spots in the *cII* gene at position #s1-5 (Yoon et al, 2010b). All these hot spots remain in Rev3 depleted cells, and Polλ depletion generates the same spectrum of hot spot mutations (Fig 3B). The persistence of all these hot spots in Rev3 or Polλ depleted cells is consistent with the requirement of Polλ/Polζ for error-free TLS through (6-4) PPs.

### Requirement of Polλ for UV-induced assembly of Polζ into replication foci

The requirement of Polλ as a scaffolding component of Polζ suggests that Polλ is targeted to RFs stalled at DNA lesion sites in conjunction with Polζ; hence, the assembly of both proteins at lesion sites would be interdependent. To examine this, we expressed Flag-Polλ in HFs and analyzed the frequency of Flag-Polλ foci-containing cells. We find that only ~3% of unirradiated cells contain Flag-Polλ foci, whereas upon UV irradiation ~15% of cells contain Flag-Polλ foci (Fig 4A). Our observation that the UV-induced increase in the frequency of Flag-Polλ foci-containing cells does not occur if Rev3 or Rev7 are depleted, indicates that Polζ is required for Polλ assembly into foci (Fig 4A). Conversely, the result that the UV-induced increase in the frequency of Rev7 foci-containing cells is ablated upon Polλ depletion conforms with the requirement of Polλ for Polζ assembly into foci (Fig 4B). The inter-dependence of Polλ and Polζ for foci formation suggests that Polλ and Polζ exist together in a physical complex in human cells and that this complex assembles at RFs stalled at UV lesions. As expected from the role of Polλ/Polζ in promoting TLS through UV lesions via pathways that operate independently of Rev1/Polη-dependent TLS, depletion of Polη or Rev1 had no effect on UV-induced assembly of Polλ into foci (Fig 4A); and conversely, depletion of Polλ conferred no impairment in UV-induced assembly of Polη into foci (Fig 4C).

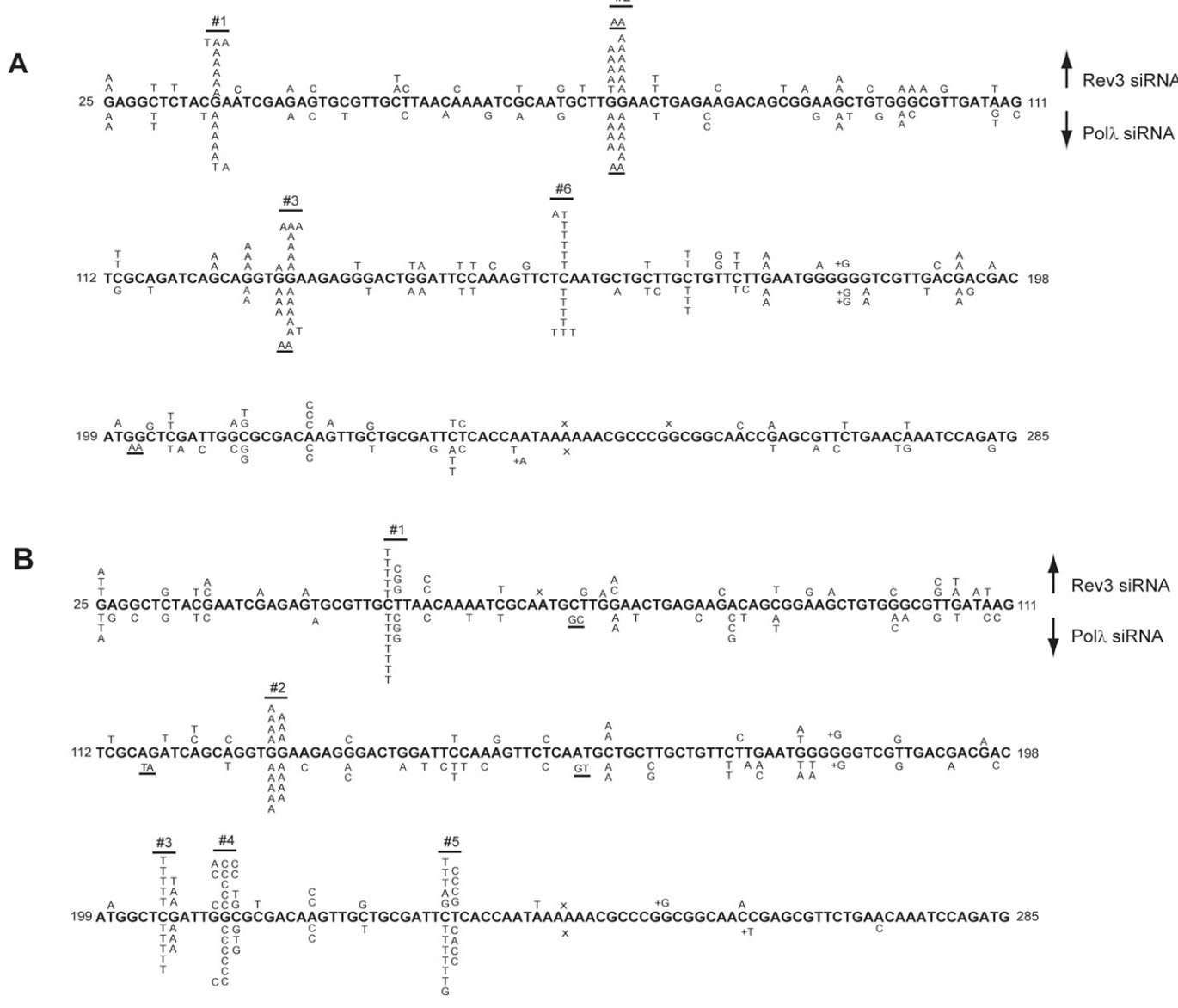

**Figure 3. Effects of Polλ depletion on UV-induced mutational spectra in the *cII* gene in BBMEFs.**
**(A)** UV-induced (5 J/m²) mutational spectra resulting from TLS through cyclobutane pyrimidine dimers (CPDs) in BBMEFs expressing (6-4) PP photolyase. Mutational spectra in Rev3-depleted cells are shown above the sequence and in Polλ depleted cells are shown below the sequence. Whereas TLS through CPDs in WT cells generates hot spots at position #s 1–11 in the *cII* gene, only hot spots at position #s 1, 2, 3, and 6 remain in Rev3- or Polλ-depleted cells. **(B)** UV-induced (5 J/m²) mutational spectra resulting from TLS through (6-4) PPs in BBMEF cells expressing CPD photolyase. Mutational spectra in Rev3 depleted cells are shown above the sequence and in Polλ depleted cells are shown below the sequence. The positions of UV-induced hot spots #1–5 are indicated.

## Physical interaction of Polλ with the Rev7 subunit of Polζ

Polλ is a 575-residue polypeptide comprised of an N-terminal BRCT domain followed by a proline-rich region and a C-terminal domain that has DNA polymerase activity (Fig 5A). In the Rev1-Polζ complex, the C-terminal region of Rev1 physically interacts with the Rev7 subunit of Polζ (Murakumo et al, 2001; Kikuchi et al, 2012). To determine whether Polλ also interacts physically with Rev7, we carried out pull-down assays in which full-length Polλ was incubated with GST-Rev7 conjugated to glutathione sepharose beads. The observation that Polλ co-elutes with GST-Rev7 indicated that the two proteins interact physically (Fig 5B). Although Polλ's BRCT domain is essential for its role in non-homologous end joining (Lee et al, 2004; Nick McElhinny et al, 2005), this domain is not required for physical interaction of Polλ with Rev7, as N-terminally deleted Polλ-NTD co-elutes with GST-Rev7 (Fig 5B). The specificity of Polλ interaction with Rev7 was confirmed by ascertaining the lack of Polλ binding to GST alone (Fig 5B).

Our results that in chromatin fractions isolated from UV-irradiated HFs, Polλ co-immunoprecipitates (co-IPs) with Rev7 and that the level of their interaction increases in response to UV irradiation add further evidence that Polλ assembles with Polζ in human cells (Fig 5C). Furthermore, in

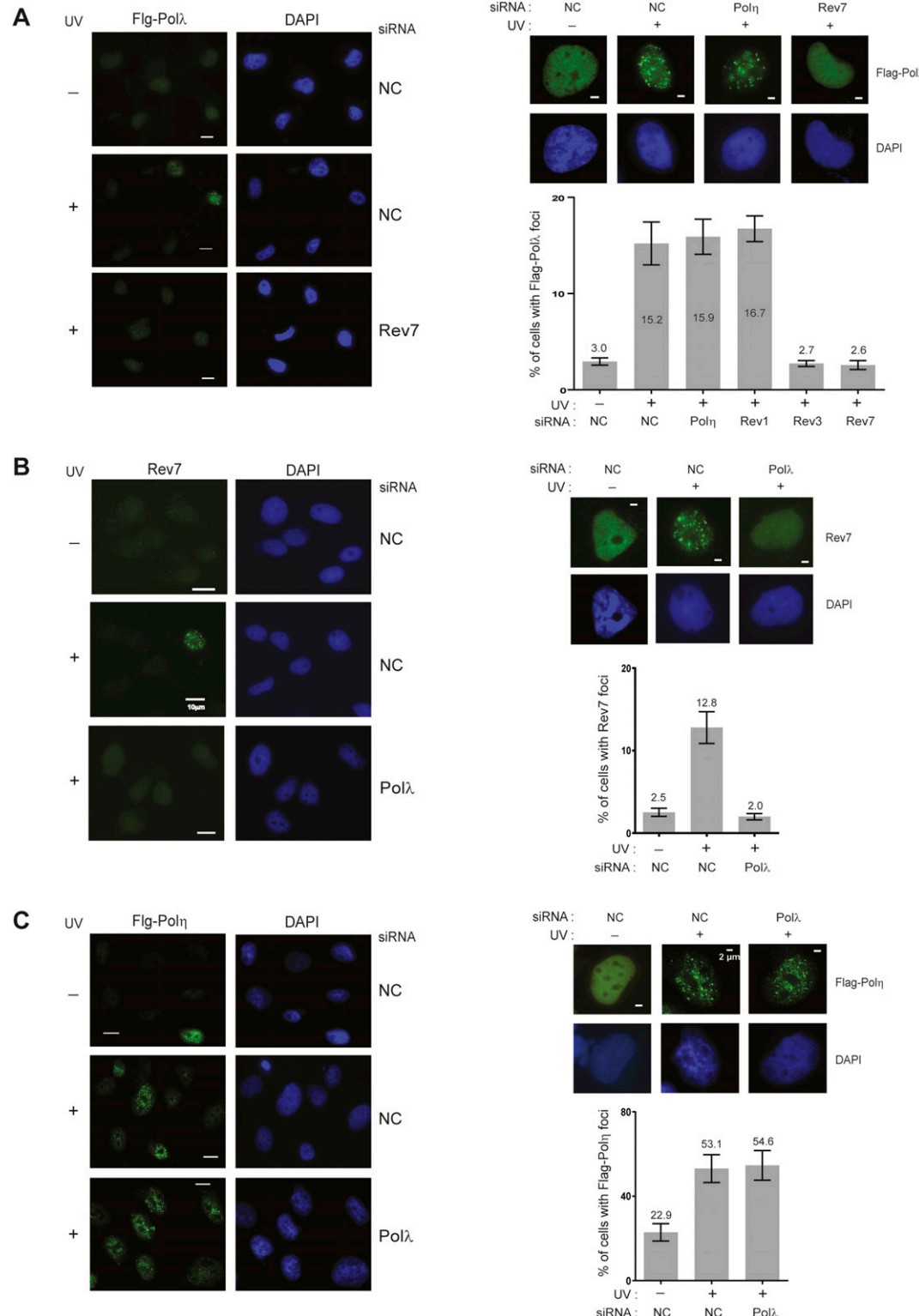

**Figure 4.   Interdependence of UV-induced assembly of Polζ and Polλ into replication foci.**
**(A)** Requirement of Rev3 or Rev7 for UV-induced assembly of Polλ into replication foci. GM637 HFs expressing Flag-Polλ were treated with siRNAs. After 48 h of siRNA treatment, cells were irradiated with UVC (30 J/m²). Cells were fixed 4 h after UV treatment, and Flag-Polλ foci were visualized by immuno-fluorescence microscopy. Representative images of multiple cells are shown on the left and magnified single cell images of Flag-Polλ foci are shown on the right on top; quantification of cells containing these foci is shown below. **(B)** Requirement of Polλ for UV-induced assembly of Rev7 into replication foci. GM637 HFs were treated with siRNAs. Rev7 foci in multiple cells are shown on the left and magnified single cell images of Rev7 foci and quantification analysis are shown on the right. **(C)** Polλ does not affect UV-induced assembly of Polη into replication foci. XPV HFs expressing Flag-Polη were treated with siRNAs. Flag-Polη foci in multiple cells are shown on the left and magnified single cell images of Flag-Polη foci and quantification analysis are shown on the right. 60× magnification was used for images. For each analysis, ~500 cells were analyzed from three independent experiments. Scale bar for foci on the left is 10 μm and for foci on the right is 2 μm.

chromatin fractions from UV-irradiated HFs, both the Polλ and Rev7 proteins co-IP with ub-PCNA, confirming that this protein assembly occurs in conjunction with ub-PCNA (Fig 5C)—a prerequisite for DNA damage–induced assembly of TLS Pols in human cells (Yoon et al, 2019b, 2015).

### Polλ's BRCT domain is not required for TLS through a (6-4) photoproduct in human cells

Our results that N-terminally deleted (245–575) Polλ physically interacts with the Rev7 subunit of Polζ suggested that Polλ's BRCT domain may not be required for its role in TLS in human cells. Because Polλ's DNA synthesis activity is required for TLS through (6-4) PPs in human cells and because TLS through the (6-4) PPs would require the sequential roles of Polλ and Polζ in nt insertion opposite the (6-4) PP and in the subsequent extension of synthesis, respectively, we examined the effects of N-terminally deleted Polλ on TLS opposite (6-4) TT PP in human cells. As shown in Table 6, in WT HFs depleted for Polλ and harboring the vector plasmid, TLS occurs with a frequency of ~13%, whereas expression of WT Polλ raises TLS frequency to ~27% and expression of catalytically inactive Polλ reduces TLS frequency to the same level as in the vector control–consistent with the requirement of Polλ DNA polymerase activity. Importantly, expression of (245–575) Polλ (Fig S1C) restores TLS frequency to WT levels (Table 6), indicating that the lack of the N-terminal domain, which includes the BRCT domain and the proline-rich region, has no untoward effect on the capacity of Polλ to carry out its role in DNA synthesis opposite (6-4) PPs as well as its scaffolding role with Polζ.

### Polλ's role in Polζ-dependent TLS contributes to UV survival

The requirement of Polλ for Polζ-dependent TLS opposite UV lesions and for Polζ localization into foci predicted that the two Pols would contribute to UV survival to the same extent. Accordingly, we find that depletion of either Polλ, Rev3, or Rev7 confers the same

level of reduction in UV survival in HFs (Fig 6A). As expected from the involvement of Polλ and Polζ in the same TLS pathway, UV sensitivity remains the same in cells co-depleted for Polλ and Rev3 as in cells depleted for Polλ or Rev3 alone (Fig 6A). Depletion of Polθ or Polη results in a greater reduction in UV survival, concordant with their relatively more significant contribution to TLS opposite UV lesions (Fig 6A). And consistent with the role of Polλ/Polζ as an alternate to Polη-dependent TLS through CPDs and (6-4) photoproducts, depletion of Polλ, Rev3, or Rev7 reduced UV survival of XPV HFs to nearly the same extent (Fig 6B). In Polλ⁻/⁻ MEFs, we confirmed that Rev3 depletion caused no further reduction in UV survival whereas depletion of Polη or Polθ reduced UV survival (Fig 6C), and as expected from their relative contribution to TLS, Polη depletion conferred a greater reduction in UV survival than Polθ depletion (Fig 6C).

### Requirement of Polλ for Polζ-dependent TLS opposite other DNA lesions

To establish that Polλ's role in TLS as an indispensable component of Polζ extends to other DNA lesions, we analyzed its role in TLS opposite thymine glycol (Tg) and 1,N⁶-ethenodeoxyadenine (εdA). Tg is generated from the reaction of thymine with hydroxyl radicals resulting from aerobic respiration and from exposure to chemical oxidants or ionizing radiation. εdA is formed in DNA through interaction with aldehydes derived from lipid peroxidation, a normal chain reaction process that initiates from the oxidation of poly-unsaturated fatty acids in cell membranes and results in the formation of a variety of highly reactive aldehydes.

We have shown previously that replication through the Tg adduct is conducted via two alternate TLS pathways dependent upon either Polκ/Polζ or Polθ (Yoon et al, 2010a, 2014) (see also Fig 7A). In the Polκ/Polζ pathway, following nt insertion by Polκ opposite the Tg adduct, Polζ would extend synthesis from the nt opposite Tg, and this pathway promotes error-free TLS through the lesion. In the

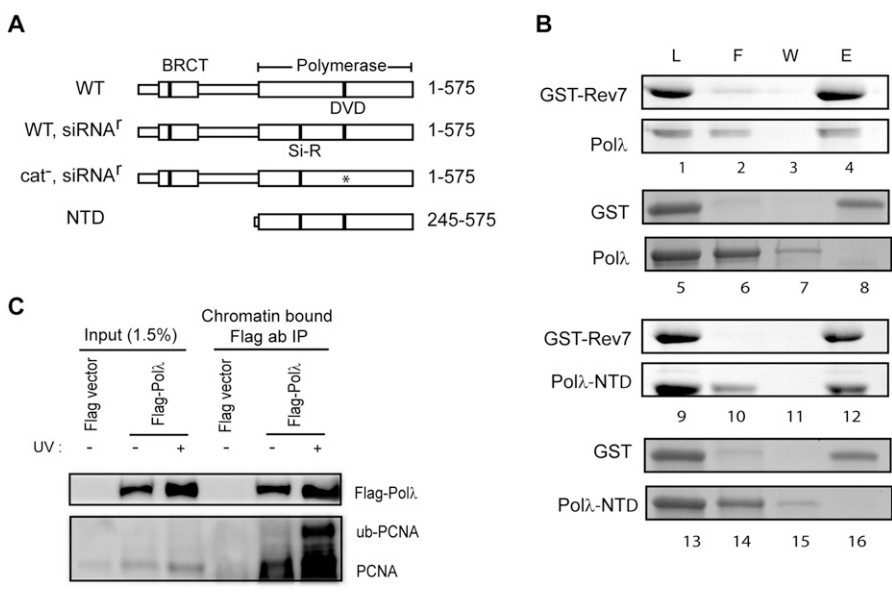

**Figure 5.  Physical interaction of Polλ with Rev7.**
**(A)** Schematic representation of the Polλ protein. Linear boxes represent the amino acid primary structure of Polλ and protein residues are given on the right. The BRCT and polymerase subdomains are labeled and indicated by larger boxes. The position of the catalytic residues Asp427 and Asp429 are indicted by DVD, and the asterisk indicates that the catalytic residues were mutated to alanines. The position of the silent siRNA resistant mutant sequence is indicted by Si-R. Genotypes are given on the left. **(B)** Physical interaction of Polλ with Rev7. Purified GST-tagged Rev7 or GST protein alone was incubated with WT Polλ or N-terminally deleted Polλ-NTD and GST affinity pull down assays were performed as described in the Materials and Methods section. Protein aliquots from the load (L), flow through (F), wash (W), and elution (E) steps were resolved by 12% SDS–PAGE gel and proteins were visualized by Coomassie blue R-250 staining. Protein identities are given on the left. **(C)** Co-immunoprecipitation (co-IP) of Flag-Polλ with Rev7 in chromatin fractions. GM637 cells expressing Flag-Polλ were UV irradiated and incubated for 4 h. Chromatin extracts were prepared and immunoprecipitated with Flag M2-agarose. Co-IP of Flag-Polλ with ub-PCNA and Rev7 was determined by Western blot analysis.

**Table 6.  Effects of WT (1–575), catalytically inactive D427A, D429A, or N-terminally deleted (245–575) Polλ on TLS opposite a (6-4) TT photoproduct carried on the leading strand in WT HFs.**

| DNA lesion | Vector expressing | Number of *Kan*[+] colonies | Number of blue colonies among *Kan*[+] | TLS (%)[a] |
|---|---|---|---|---|
| (6-4) TT PP | Vector control | 346 | 45 | 13.0 |
| | WT (1–575) Polλ | 438 | 119 | 27.2 |
| | D427A, D429A Polλ | 418 | 53 | 12.7 |
| | (245–575) Polλ | 396 | 101 | 25.5 |

[a]Statistical analyses of these data are shown in Table S4.

alternative pathway, Polθ performs both steps of TLS and it acts in an error-prone manner (Yoon et al, 2010a, 2014). As shown in Table 7, TLS opposite Tg occurs at a frequency of ~23% in WT MEFs. In Polλ$^{-/-}$ MEFs, TLS is reduced to ~10% and a similar reduction in TLS frequency occurs upon depletion of either Polκ or Rev3, consistent with a role of Polλ in Polζ-dependent TLS in the Polκ/Polζ pathway. In Polλ$^{-/-}$ MEFs depleted for Polθ, or in Polθ$^{-/-}$ MEFs depleted for Polλ, where both the Polκ/Polζ and Polθ pathways would be rendered inactive, TLS is almost completely abrogated, as only a very low residual level of TLS (~1%) remains. From these data, we infer a role for Polλ in conjunction with Polζ in Polκ/Polζ-dependent TLS pathway opposite Tg (Fig 7A).

TLS opposite εdA is performed by a major Polι/Polζ-dependent pathway in which after nt insertion opposite εdA by Polι, Polζ would extend synthesis, and this pathway operates in an error-free manner (Yoon et al, 2019a) (see also Fig 7B). In the Polι/Polζ pathway, Rev1 performs a non-catalytic role as a scaffolding component of Polι. In another relatively minor pathway, Rev1 functions in TLS opposite εdA as a DNA polymerase and acts in an error-prone manner. Polθ promotes TLS through εdA via a third pathway and it also operates in an error-prone manner (Yoon et al, 2019a). As shown in Table 7, TLS opposite εdA occurs at a frequency of ~22% in WT MEFs. TLS frequency is reduced to ~10% in Polλ$^{-/-}$ MEFs and TLS frequency remains the same in Polλ$^{-/-}$ MEFs depleted for Polι or Rev3 (Table 7), consistent with the requirement of Polλ for Polζ-dependent TLS in the Polι/Polζ pathway. In Polλ$^{-/-}$ MEFs depleted for Polθ or in Polθ$^{-/-}$ MEFs depleted for Polλ or Rev3 (Table 7), only residual TLS (~3%) remains; it reflects the minor contribution of Rev1 polymerase-dependent TLS that operates in the absence of the Polι/Polζ and Polθ pathways (Yoon et al, 2019a). These data in Polλ$^{-/-}$ and Polθ$^{-/-}$ MEFs conform with a role of Polλ in conjunction with Polζ in Polι/Polζ-dependent TLS opposite εdA (Fig 7B).

### Polλ polymerase activity is not required for TLS opposite Tg and εdA adducts

Our proposal that after nt insertion by Polκ opposite the Tg lesion, Polζ would extend synthesis (Yoon et al, 2010a) predicted that only the non-catalytic scaffolding role of Polλ would be required for Polζ-dependent TLS opposite Tg. Similarly, the role of Polι in inserting a nt opposite εdA and of Polζ in extending subsequent synthesis (Nair et al, 2006; Yoon et al, 2019a) predicted that only the non-catalytic scaffolding role of Polλ would be required for Polζ-dependent TLS opposite εdA. To verify these predictions, we expressed WT Polλ and DNA polymerase–defective D427A, D429A

mutant Polλ in Polλ$^{-/-}$ MEFs and analyzed the frequency of TLS opposite the Tg or εdA lesion carried on the duplex plasmid in these MEFs. In Polλ$^{-/-}$ MEFs, TLS opposite Tg occurs with a frequency of ~10% (Table 3). Our results that TLS frequency in Polλ$^{-/-}$ MEFs expressing WT Polλ rises to ~21% and that TLS frequency in Polλ$^{-/-}$ MEFs expressing catalytically inactive mutant Polλ remains the same as in Polλ$^{-/-}$ MEFs expressing WT Polλ confirm that only the non-catalytic scaffolding role of Polλ is required for TLS opposite Tg (Table 3 and Fig 7A). And the observation that catalytically inactive Polλ supports WT levels of TLS opposite εdA indicates that only the non-catalytic scaffolding role of Polλ is required for TLS opposite this DNA lesion (Table 3 and Fig 7B).

### Requirement of Polζ DNA polymerase activity for TLS through (6-4) TT photoproduct

The biochemical evidence that purified Polλ can insert a nt opposite both the 3′T and 5′T of the (6-4) TT PP (Fig 2B) raised the possibility that Polζ's DNA polymerase activity may not be required for TLS through this DNA lesion. To examine this, we expressed full-length WT Rev3 and mutant Rev3 in which the active site residues D2781 and D2783 have both been changed to alanines. In HFs depleted for Rev3 and expressing no Rev3 (vector control), TLS opposite (6-4) TT PP occurs at a frequency of ~14%. TLS frequency rises to 27% in cells expressing WT Rev3, whereas in cells expressing the D2781A and D2783A Rev3 mutant, TLS frequency is reduced to the same level as in cells harboring the vector control (Table 8). Thus, Rev3 DNA polymerase activity is required for TLS through (6-4) TT PP.

## Discussion

### Polλ promotes replication through DNA lesions in conjunction with Polζ

Genetic and biochemical studies have established an indispensable role of Rev1 as a scaffolding component of Polζ in yeast and Rev1/Polζ-dependent TLS operates in a highly error-prone manner (Baynton et al, 1999; Nelson et al, 2000; Haracska et al, 2001; Gibbs et al, 2005; Acharya et al, 2006). Moreover, Rev1 functions together with Polζ in TLS that occurs during gap filling reactions in human or mouse cells (Yoon et al, 2015) and the dependence of cancer cells upon Rev1/Polζ for DNA damage induced mutagenesis (Doles et al, 2010; Xie et al, 2010) suggests that during malignant transformation, cancer cells acquire the potential of highly elevated mutability, afforded by Rev1/Polζ. In striking contrast, TLS analyses with a

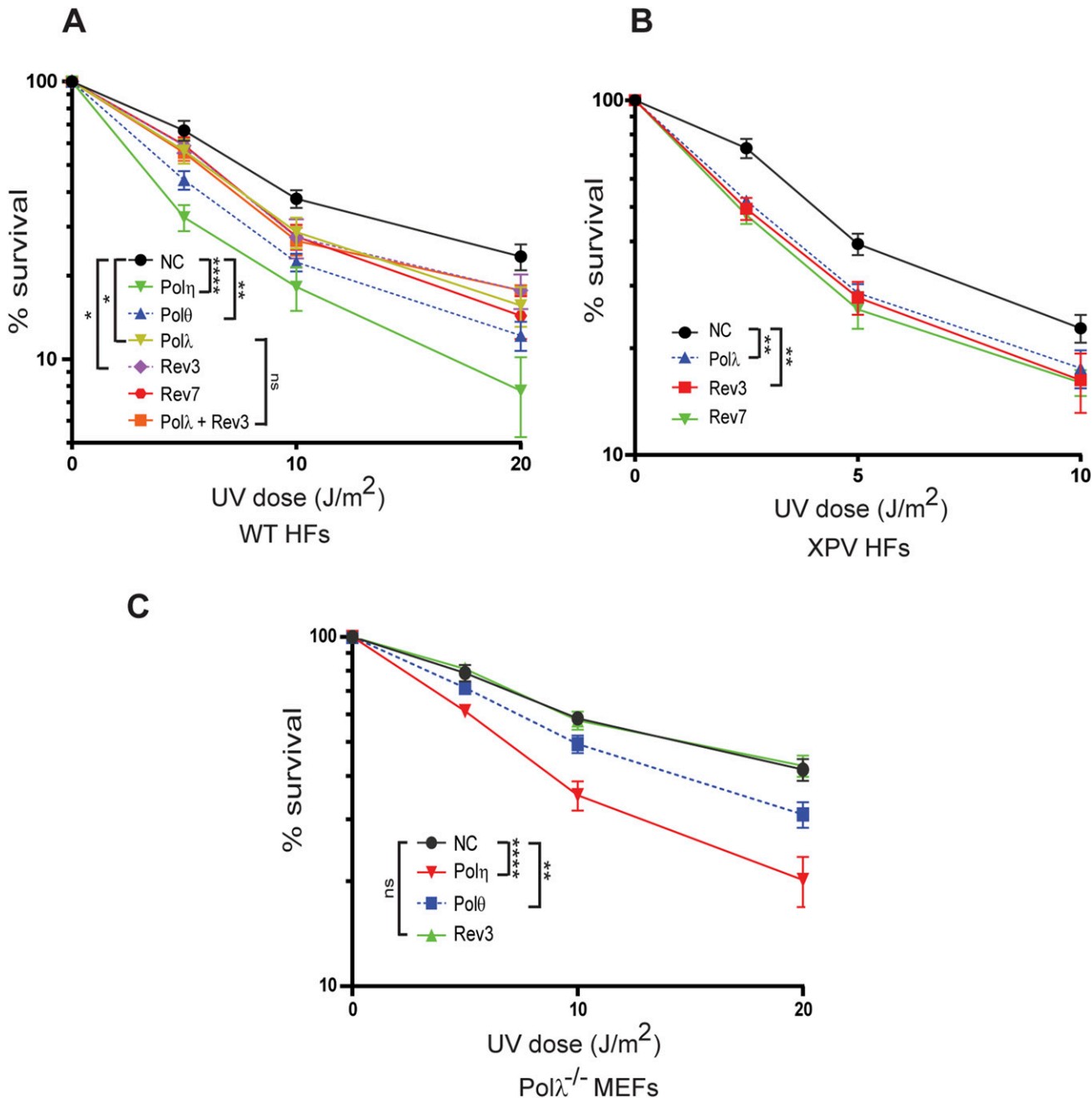

**Figure 6. UV survival assays.**
GM637 HFs, XPV (XP30R0) HFs, or Polλ$^{-/-}$ MEFs were treated with siRNAs for 48 h and irradiated with UV light in PBS buffer. Cells were incubated for additional 48 h after UV irradiation and UV survival was determined by the MTS assay. The data represent the mean and SD of results of four independent experiments. Student's two-tailed $t$ test $P$-values; ns, not significant; *$P < 0.05$; **$P < 0.01$; ****$P < 0.0001$. **(A)** UV cytotoxicity of GM637 HFs depleted for Polλ, Polζ, or other TLS Pols. **(B)** UV cytotoxicity of XPV HFs depleted for Polλ or Polζ. **(C)** UV cytotoxicity of Polλ$^{-/-}$ MEFs depleted for Polζ, Polη, or Polθ.

variety of DNA lesions present on the template for leading or lagging strand replication of a duplex plasmid carried in normal human cells (not derived from cancers) have shown that Rev1 functions together with Y-family Pols η, ι, or κ and not with ζ (Yoon et al, 2019a, 2018, 2017, 2015). Importantly, the TLS mechanisms inferred from the duplex plasmid system reflect the mechanisms that operate during cellular replication in normal cells. For example, the reduction in the rate of RF progression through UV lesions in

Polη$^{-/-}$, Polθ$^{-/-}$, and Polη$^{-/-}$ Polθ$^{-/-}$ primary MEFs conforms with the roles deduced for these Pols from TLS analyses opposite UV lesions in the plasmid system (Yoon et al, 2019b). Altogether, the evidence indicates that the TLS mechanisms that operate during replication in normal human cells differ from the mechanisms inferred from gap repair assays or from studies in cancer cells.

The requirement of Rev1 for TLS mediated by Y-family Pols and not for Polζ-dependent TLS suggested that some other protein

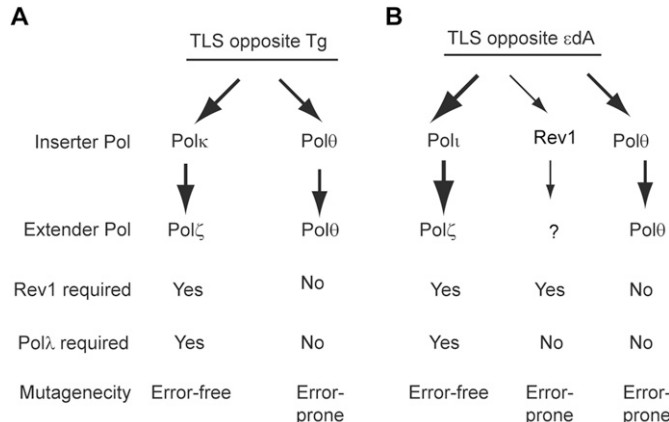

**A** TLS opposite Tg

**B** TLS opposite εdA

| | | | | | | |
|---|---|---|---|---|---|---|
| Inserter Pol | Polκ | Polθ | Polι | Rev1 | Polθ |
| Extender Pol | Polζ | Polθ | Polζ | ? | Polθ |
| Rev1 required | Yes | No | Yes | Yes | No |
| Polλ required | Yes | No | Yes | No | No |
| Mutagenecity | Error-free | Error-prone | Error-free | Error-prone | Error-prone |

**Figure 7. TLS pathways for replication through Tg and εdA lesions.**
**(A)** TLS pathways for replication through Tg. After nt insertion by Polκ, Polζ extends synthesis. Polλ's scaffolding role is required for Polζ function in TLS. This pathway conducts error-free TLS through the Tg lesion. The alternative Polθ-dependent TLS operates in an error-prone manner. **(B)** TLS pathways for replication through εdA. After nt insertion by Polι, Polζ extends synthesis. Polλ's scaffolding role is required for Polζ function in TLS. This pathway operates in an error-free manner. Polθ-dependent TLS provides the other major TLS pathway and TLS requiring Rev1 polymerase function contributes to a relatively minor pathway. Polθ and Rev1 polymerase–dependent TLS operate in an error-prone manner. The thickness of the arrows depicts the relative contribution of TLS Pols to lesion bypass.

might function in lieu of Rev1 with Polζ. Here, we provide evidence that Polλ functions in TLS in conjunction with Polζ, which include the following: (1) For TLS opposite a *cis-syn* TT dimer, Polλ functions together with Polζ in an error-prone pathway dependent upon the sequential action of Polθ and Polλ/Polζ. (2) For TLS opposite a (6-4) TT photoproduct, Polλ functions together with Polζ in an error-free TLS pathway that provides an alternative to the error-prone TLS mediated by the sequential action of Polι with Polθ or of Polη with

Polθ. (3) From analyses of UV-induced mutations opposite CPDs and (6-4) PPs analyzed in the *cII* gene in BBMEFs, we established that Polλ affects mutation frequencies and mutational spectra similar to that by Polζ. (4) UV-induced accumulation of Polζ into replication foci requires Polλ and vice versa, and Polλ physically interacts with the Rev7 subunit of Polζ. (5) In HFs, Polλ depletion reduces UV survival to the same extent as occurs upon Polζ depletion, and co-depletion of Polλ with Polζ causes no further reduction in UV survival. And Polζ depletion causes no further reduction in UV survival in Polλ$^{-/-}$ MEFs. (6) Opposite DNA lesions that include Tg and εdA adducts, Polλ functions together with Polζ in error-free TLS pathways.

## Role of Polλ as a scaffolding component of Polζ

Our evidence that Polλ's DNA polymerase activity is not required for TLS opposite CPDs, Tg, or εdA indicates that only the scaffolding role of Polλ is required for Polζ function in TLS opposite these DNA lesions. The lack of requirement of Polλ polymerase activity for TLS opposite these DNA lesions is compatible with the roles that have been deduced from biochemical and structural studies and assigned to other Pols. The requirement of Polλ DNA polymerase activity for TLS opposite (6-4) TT PPs in human cells and the ability of purified Polλ to insert a nt opposite the 3'T and 5'T of the (6-4) TT PP supports a role for Polλ in nt insertion opposite the photoproduct from which Polζ would extend synthesis. Thus, in addition to its indispensable non-catalytic role with Polζ, Polλ polymerase activity is required for TLS opposite (6-4) PPs.

## Nucleotide insertion opposite (6-4) photoproduct by Polλ

The evidence that purified Polλ inserts an A opposite both the 3'T and 5'T residue of (6-4) TT conforms with the genetic evidence that Polλ/Polζ-dependent TLS operates in an error-free manner in

**Table 7. The effects of siRNA knockdowns of TLS Pols on replicative bypass of a thymine glycol or a 1, N⁶-ethenodeoxyadenosine carried on the leading strand template in wild type, Polλ$^{-/-}$, or Polθ$^{-/-}$ MEFs.**

| DNA lesion | MEFs | siRNA | Number of *Kan⁺* colonies | Number of blue colonies among *Kan⁺* colonies | TLS (%)[a] |
|---|---|---|---|---|---|
| Tg | WT | NC | 317 | 73 | 23.0 |
| | Polλ$^{-/-}$ | NC | 405 | 42 | 10.4 |
| | | Polκ | 381 | 41 | 10.8 |
| | | Rev3 | 423 | 48 | 11.3 |
| | | Polθ | 462 | 6 | 1.3 |
| | Polθ$^{-/-}$ | NC | 360 | 42 | 11.7 |
| | | Polλ | 428 | 6 | 1.4 |
| εdA | WT | NC | 322 | 70 | 21.7 |
| | Polλ$^{-/-}$ | NC | 304 | 32 | 10.5 |
| | | Polι | 337 | 38 | 11.3 |
| | | Rev3 | 388 | 42 | 10.8 |
| | | Polθ | 334 | 11 | 3.3 |
| | Polθ$^{-/-}$ | NC | 341 | 36 | 10.6 |
| | | Polλ | 401 | 13 | 3.2 |
| | | Rev3 | 312 | 12 | 3.8 |

[a]Statistical analyses of these data are shown in Table S5.

**Table 8. Effects of catalytically inactive D2781A and D2783A Rev3 on TLS opposite a (6-4) TT photoproduct carried on the leading strand in WT HFs.**

| DNA lesion | Vector expressing | Number of *Kan*+ colonies | Number of blue colonies among *Kan*+ | TLS (%)[a] |
|---|---|---|---|---|
| (6-4) TT PP | Vector control | 408 | 58 | 14.2 |
| | WT (1–3130) Rev3 | 318 | 86 | 27.0 |
| | D2781A and D2783A Rev3 | 341 | 47 | 13.8 |

[a]Statistical analyses of these data are shown in Table S6.

human cells. A (6-4) TT PP induces a large structural distortion in DNA in which it confers a 44° bend in the DNA helix and the 3′T is oriented perpendicular to the 5′T (Kemmink et al, 1987; Kim & Choi, 1995; Kim et al, 1995; Lee et al, 1999), impairing the ability of the 3′T to form a normal W-C pair with the correct nt. Since Polλ normally uses W-C base pairing for DNA synthesis, Polλ may not accommodate a (6-4) PP in its active site such that the 3′ pyrimidine residue of the photoproduct forms a W-C base pair with the correct incoming nt. That raises the possibility that purified Polλ inserts an A opposite both pyrimidines of the photoproduct, wherein the lesion adopts an extrahelical configuration resembling an abasic lesion.

Regardless of whether purified Polλ performs error-free nt incorporation opposite (6-4) TT photoproduct by W-C base pairing or via an "abasic-like" mode, our genetic evidence that Polλ DNA polymerase activity is required for TLS opposite (6-4) PPs (Table 3) and that Polλ conducts error-free TLS opposite (6-4) PPs formed at CC, TC, or TT sequences in the *cII* gene integrated into the genome (Table 5 and Fig 3B) implies that in human and mouse cells, Polλ incorporates a correct nt opposite the (6-4) photoproducts formed at dipyrimidine sequences from which Polζ extends synthesis.

### Implications of Polλ requirement for Polζ function in TLS in human cells

Our TLS analyses have indicated that, overall, Polλ/Polζ-dependent TLS operates in an error-free fashion in human cells. Thus, of the four DNA lesions examined here, Polλ/Polζ contributes to error-free TLS opposite (6-4) PPs, Tg, and εdA, whereas it functions in an error-prone TLS pathway opposite CPDs. The ability of Polλ/Polζ to perform error-free TLS through (6-4) PPs stands in stark contrast to the role of Rev1/Polζ in error-prone TLS opposite (6-4) PPs in yeast cells (Gibbs et al, 2005) and the role of Rev1/Polζ in error-prone TLS opposite (6-4) TT PP that occurs during gap filling reaction in human or mouse cells (Yoon et al, 2015). The requirement of Polλ/Polζ for error-free TLS through the Tg and εdA lesions further suggests that after the insertion of a nt by Polκ opposite Tg, or the insertion of a nt by Polι opposite εdA, Polζ in combination with Polλ performs error-free synthesis during the extension phase of TLS. In the Polλ/Polζ ensemble, Polζ may be inhibited from extending synthesis from the wrong nts inserted opposite DNA lesions by the other Pols, promoting thereby a more error-free mode of TLS. In this regard, Polλ's effect on TLS by Polζ would be diametrically opposite to the enhancement in the proficiency of extension of synthesis from the wrong nts that occurs in TLS mediated by Rev1/Polζ (Acharya et al, 2006).

How might Polλ affect Polζ function in TLS? Our evidence that UV-induced accumulation of Polζ or Polλ into replication foci depends upon each other suggests that the assembly of Polλ together with Polζ into foci requires both Pols. In addition to its association with Polζ,

Polλ may promote the assembly of additional proteins which serve to increase the fidelity and efficiency of the Polλ/Polζ ensemble in TLS. In the Polλ/Polζ multi-protein ensemble, Polλ may acquire the ability to form W-C base pair with the correct nt opposite the (6-4) PPs formed at the TT, TC, or CC dipyrimidine sites. In thymine glycol, because of the addition of hydroxyl groups at C5 and C6, the C5 methyl group protrudes in the axial direction and that prevents the base 5′ to Tg from stacking above it. Hence, the insertion of a correct nt opposite this extra-helical base would require Polζ to accommodate this base in its active site such that it engages in W-C pairing with the correct nt. In the multi-protein ensemble of Polζ with Polλ and other proteins, the extra-helical base may be pushed into a normal helical configuration in the Polζ active site enabling Polζ to carry out nt insertion opposite the base 5′ to Tg with a high fidelity. Similarly, any structural distortions in the DNA helix imposed by the εdA lesion at the extension step of TLS could be minimized in the Polζ active site in the Polλ/Polζ ensemble. Altogether, the adoption of Polλ as an integral component of Polζ may have evolved to provide human cells a strategy for imposing a predominantly error-free mode of TLS upon Polζ.

### Conclusions

Our evidence that Polλ functions as an indispensable scaffolding component of Polζ for TLS that occurs during replication in normal human cells depicts an entirely new and unexpected role for Polλ. Although similar to Rev1 in its scaffolding role, Polλ affects Polζ-dependent TLS differently than Rev1. Thus, unlike the role of Rev1/Polζ in error-prone TLS opposite (6-4) photoproduct, Polλ, by incorporating a correct nt opposite the photoproduct by W-C base pairing, would enable Polλ/Polζ-dependent TLS to operate in an error-free manner in human cells. This role of Polλ in TLS opposite (6-4) photoproduct strongly suggests that opposite many other such distorting DNA lesions, Polλ would similarly insert a correct nt from which Polζ would extend synthesis. Similar to the role of Polλ/Polζ in error-free TLS opposite the Tg and εdA lesion, complex formation with Polλ may enable Polζ to operate in an error-free manner opposite other DNA lesions where only its scaffolding role is required.

# Materials and Methods

### Expression and purification of Polλ

The cDNA's encoding the WT Polλ (residues 1–575) or the N-terminally truncated Polλ (residues 245–575) were each cloned in frame with a PreScission Protease cleavable GST tag in pBJ842 (Johnson et al, 2006) generating plasmids pBJ2043 and pBJ1431, respectively. Plasmids were transformed into yeast strain YRP654 and protein expression was induced

by the addition of 2% galactose. Proteins were purified using a standard GST purification protocol (Johnson et al, 2006). Briefly, frozen yeast cells were resuspended in cell breakage buffer (50 mM Tris-HCl, pH 7.5, 20% sucrose, 500 mM NaCl, 1 mM EDTA, 10 mM $\beta$-mercaptoethanol, 1 mM benzamidine HCl, and complete-EDTA-free protease inhibitors [Roche]) and disrupted on a French press. Clarified protein extract was treated with 0.208 g/ml ammonium sulfate, and precipitated proteins were resuspended and dialyzed overnight in glutathione binding buffer (GBB, 50 mM Tris-HCl, pH 7.5, 10% glycerol, 500 mM NaCl, 1 mM benzamidine HCl, and Roche complete protease inhibitors). Proteins were incubated with glutathione sepharose for several hours before washing with GBB containing 1 M NaCl. Sepharose was then equilibrated in GBB containing 150 mM NaCl without PI's, and proteins were eluted by treatment with PreScission Protease (GE Healthcare) overnight at 4°C. Proteins were harvested and residual PreScission Protease was removed by incubation with fresh Glutathione Sepharose. Proteins were concentrated and aliquots frozen at −70°C.

## DNA polymerase assays

The standard DNA polymerase reaction (5 $\mu$l) contained 25 mM Tris–HCl (pH 7.5), 5 mM MgCl$_2$, 1 mM DTT, 10% glycerol, 100 $\mu$g/ml BSA, and 5 nM DNA substrate. Reactions contained 100 $\mu$M of either a single deoxynucleotide triphosphate (dGTP, dATP, dTTP, or dCTP) or 100 $\mu$M each of all four. The DNA substrate was generated by annealing a 5′ $^{32}$P labeled oligonucleotide primer to a 75-nucleotide oligonucleotide template of the sequence 5′-AGCAAGTCAC CAATGTCTAA GAGTT CGTA**T-T**ATGC CTACACTGGA GTACCGGAGC TACGTCGTGA CTGGGAAAAC-3′, which was either unmodified or contained a (6-4) TT photoproduct or a *cis-syn* TT dimer (CPD) at the underlined position. The *cis-syn* TT dimer and the (6-4) TT photoproducts were introduced into the 10mer 5′-CGTATTATGC-3′ by UV treatment, and following HPLC purification, ligated to flanking 25-mer and 40-mer oligonucleotides as described (Johnson et al, 2001) to generate the 75-mer. For standing start reactions, the 44 nucleotide primer 5′-GTTTTCCCAG TCACGACGAT GCTCCGGTAC TCCAGTGTAG GCAT-3′ was annealed to the template, and for running start reactions, the 40-mer 5′-GTTTTCCCAG TCACGACGAT GCTCCGGTAC TCCAGTGTAG-3′ was used. All oligonucleotides were PAGE-purified. Reactions contained 5 nM DNA polymerase $\lambda$ and were carried out at 37°C for 20 min. The reactions were stopped by addition of 30 $\mu$l of loading buffer (95% formamide, 20 mM EDTA, 0.3% bromophenol blue, and 0.3% xylene cyanol), and the reaction products were resolved on a 10% TBE-polyacrylamide gel containing 8M urea. Gels were dried and products were visualized by phoshorimaging on a Typhoon FLA 7000 (GE Healthcare Life Sciences).

## GST pull down experiments

The physical interaction between GST-Rev7 and Pol$\lambda$ was carried out by using a protocol described before (Haracska et al, 2002) with minor modifications. Briefly, ~2 $\mu$g of either GST tagged Rev7 or GST protein each bound to 20 $\mu$L glutathione sepharose beads was incubated with an equal amount of either full-length Pol$\lambda$ or N-terminally deleted Pol$\lambda$-NTD in 50 $\mu$l buffer I (50 mM Tris-HCl [pH 7.5], 150 mM NaCl, 5 mM dithiothreitol, 0.01% Nonidet P-40, and 10% Glycerol). Protein-glutathione Sepharose mixtures were incubated at 25°C for 15 min, followed by incubation at 4°C overnight with constant rocking. The beads were harvested by centrifugation at 1,000*g* and the unbound protein flow through (F) fraction was collected. The beads were then washed three times each with 10 vol of buffer I. The

bound proteins were then eluted with 20 $\mu$l of 1× SDS–PAGE loading buffer. Aliquots of the protein mixture load (L), the flow through (F), the last washing fraction (W), and the eluted protein fraction (E) were resolved on a 12% denaturing polyacrylamide gel, followed by Coomassie Blue R-250 staining.

## Construction of plasmid vectors containing a cis-syn TT dimer or a (6-4) TT photoproduct

The heteroduplex vectors containing a *cis-syn* TT dimer or a (6-4) TT photoproduct on the leading or lagging strand template were constructed as described previously (Yoon et al, 2009, 2010a).

## Cell lines and cell culture

Normal human fibroblasts (Coriell Institute Cell Repository, GM00637), XPA-deficient fibroblasts (Coriell Institute Cell Repository, GM04429), XPV-deficient fibroblasts (Coriell Institute Cell Repository, GM03617), SV40-transformed Pol$\lambda^{-/-}$ MEFs, and big blue mouse embryonic fibroblasts (Agilent) were grown in DMEM medium (GenDEPOT) containing 10% fetal bovine serum (GenDEPOT) and 1% antibiotic-antimycotic (GenDEPOT). Cells were grown on plastic culture dishes at 37°C in a humidified incubator with 5% CO$_2$

## Translesion synthesis assays in HFs and Pol$\lambda^{-/-}$ MEFs

For siRNA knockdown of Pol$\lambda$, HPLC-purified duplex siRNA for human and mouse genes were purchased from Ambion. The siRNA sequences were human Pol$\lambda$: 5′-GCUGGACCAUAUCAGUGAG-3′; mouse Pol$\lambda$: 5′-GCACUACGAUGACUUCCUG-3′. The efficiency of Pol$\lambda$ knockdown was verified by Western blot analysis (Fig S1A). The siRNA knockdown efficiency of other TLS Pols as well as the detailed methods for TLS assay and for mutational analyses have been described previously (Yoon et al, 2015, 2010b, 2009).

## Western blot analysis

48 h after siRNA transfection, the cells were washed with PBS buffer and lysed with RIPA buffer (1× PBS, 1% IP-40, 0.5% sodium deoxycholate, and 0.1% SDS). After 1 h incubation on ice, the cellular mixture was centrifuged, and the supernatant was collected. Equivalent amounts (~30 $\mu$g) of prepared cellular extracts were separated on a 10% SDS–polyacrylamide gel and transferred to a PVDF membrane (Bio-Rad). The membranes were probed with antibodies against human Pol$\lambda$ (Bethyl Lab) or Flag (Sigma-Aldrich), followed by appropriate secondary antibodies conjugated with horseradish peroxidase. The signals were detected using ECL-Plus (GenDEPOT). For the loading control, anti-$\beta$-tubulin antibody (Santa Cruz Biotechnology) or anti-lamin B1 antibody (Abcam) was used.

## Stable expression of wild type and mutant Pol$\lambda$ in HFs and Pol$\lambda^{-/-}$ MEFs

siRNA-resistant wild-type human Pol$\lambda$, or catalytic mutant (D427A D429A) Pol$\lambda$, or N-terminally deleted (245–575) Pol$\lambda$ cDNAs were cloned into the pCMV7-3xFlag-zeo vector (Sigma-Aldrich). The vectors were each transfected into SV40-transformed GM637 HFs or Pol$\lambda^{-/-}$ MEFs

by lipofectamine 2000 reagent (Invitrogen). After 24 h incubation, 0.5 μg of zeocin (GenDEPOT) was added to the culture media. After an additional 3 d of incubation, the cells were washed with PBS buffer and were continuously cultured in media containing 250 ng of zeocin for ~2 wk. Protein expression and siRNA knockdown efficiency were verified by Western blot analysis (Fig S1A–C).

## Stable expression of wild-type and mutant Rev3 in HFs

siRNA-resistant wild-type human Rev3, or catalytic mutant (D2781A D2783A) Rev3 cDNAs were cloned into the pCMV7-3xFlag-zeo vector (Sigma-Aldrich). The vectors were each transfected into SV40 transformed GM637 HFs by lipofectamine 2000 reagent (Invitrogen). After 24 h incubation, 0.5 μg of zeocin (GenDEPOT) was added to the culture media. After an additional 3 d of incubation, the cells were washed with PBS buffer and were continuously cultured in media containing 250 ng of zeocin for ~2 wk. Protein expression was verified by Western blot analysis (Fig S1D).

## UV survival assays

GM637 HFs, XPV (XP30R0) HFs, or Pol$λ^{-/-}$ MEFs were transfected with siRNAs and 48 h after siRNA transfection, the cells were treated with UV. For UV irradiation, the cells were washed with PBS buffer and irradiated with various doses (0–20 J/m$^2$) of UVC light in the presence of PBS buffer. After irradiation, the cells were incubated in fresh growth medium for an additional 48 h. UV cytotoxicity was determined by the MTS assay (Promega). Briefly, 100 μl of MTS assay solution was added to each well and incubated for 30 min. Cell viability was determined by measuring OD at 490 nM, and four independent experiments were carried out.

## Big blue mouse cell line and siRNA knockdown

BBMEFs were grown in DMEM medium containing 10% FBS (GenDEPOT) and antibiotics. For the cII mutation assay, the cells were plated on 100-mm plates at 50% confluence (~5 × 10$^6$ cells) and 500 pmoles of synthetic duplex siRNAs were transfected using 50 μl of Lipofectamine 2000 reagent (Invitrogen) following the manufacturer's instructions.

## UV irradiation, photoreactivation, and cII mutational assays in siRNA treated BBMEFs

48 h after siRNA knockdown, the cells were washed with HBSS buffer (Invitrogen) and irradiated at 5 J/m$^2$ with UVC light, followed by photoreactivation for 3 h at room temperature as previously described (Yoon et al, 2009, 2010b). Fresh growth medium was added, and the cells were incubated for 24 h. After that, the second siRNA transfection was carried out to maintain the siRNA knockdown of the target gene(s). Cells were incubated for an additional 4 d to allow for mutation fixation. Mouse genomic DNA was isolated using the genomic DNA isolation kit (QIAGEN). The LIZ shuttle vector was rescued from the genomic DNA by mixing DNA aliquots and transpack packaging extract (Stratagene), and the cII assay was carried out as previously described (Yoon et al, 2009, 2010b). The mutation frequency was calculated by dividing the number of mutant plaques by the number of total plaques. For mutation analysis, the sequence of PCR products of the cII gene from the mutant plaques were analyzed as described previously (Yoon et al, 2009, 2010b).

## UV-induced foci formation of Polλ, Rev7, and Polη

Full-length human Polη or Polλ ORFs were cloned into the pCMV7-3xFlag-zeo vector (Sigma-Aldrich) and stably expressed into XPV HFs or GM637 HFs, respectively. For Rev7 foci formation, Rev7 antibody (BD Biosciences) was used for immunofluorescence in GM637 HFs. Cells were suspended and treated with siRNA and cultured on a coverslip in six-well plate with 50% confluence. After 48 h, cells were treated with UVC (30 J/m$^2$). For UV irradiation, cells were washed with PBS buffer and irradiated with UVC light in the presence of PBS buffer. After irradiation, fresh growth medium was added to cells and the cells were incubated for 6 h. After washing with PBS buffer, cells were fixed with 4% paraformaldehyde for 30 min. Fixed cells were permeabilized with 0.2% Triton x-100 in PBS buffer. Primary FLAG antibody (Sigma-Aldrich) or Rev7 antibody (BD Biosciences) were added to cells in PBST (0.1% Tween 20 in PBS) containing 3% BSA. Nuclear staining was performed with DAPI (Molecular Probe) in PBS buffer for 20 min. The fluorescent images were visualized and captured by fluorescence microscopy (Nikon fluorescence microscope).

## Co-immunoprecipitation of proteins in chromatin extracts

GM637 HFs stably expressing Flag-Polλ were cultured in 15 cm plates with ~80% confluence. Cells were washed with PBS buffer and irradiated with UVC (30 J/m$^2$) in the presence of PBS buffer. After UV irradiation, cells were incubated in growth media for 4 h. For chromatin bound nuclear extracts, cells were lysed with CSK (Cytoskeleton) buffer (10 mM Hepes, pH 6.8, 100 mM NaCl, 300 mM sucrose, 3 mM MgCl$_2$, 0.5% Triton X-100, and e-complete protease inhibitors), and chromatin extracts were treated with 1% formalin in PBS buffer for 10 min at room temperature followed by 125 mM glycine addition. Cell pellets were resuspended in PBS buffer containing 30 units of Xpernase (GenDEPOT). Extracts were incubated at room temperature for 10 min and then centrifuged to isolate the chromatin extracts. 2 mg of chromatin extracts were diluted with an equal volume of immunoprecipitation (IP) buffer (150 mM NaCl, Tris–HCl, pH 7.5, 1 mM EDTA, 0.05% NP40, 10% glycerol, and protease inhibitors) and mixed with 0.5 μg of FLAG agarose beads overnight at 4°C. Flag agarose beads were washed with IP buffer twice, and bound proteins were eluted in Laemmli buffer (2% SDS, 10% glycerol, 60 mM Tris–HCl, pH 6.8, 100 mM DTT, and 0.05% bromophenol blue). PCNA ab (Santa Cruz Biotechnology), Rev 7ab (BD biosciences), or Flag ab (Sigma-Aldrich) were used for Western blot analysis.

# Supplementary Information

# Acknowledgements

We thank Robert Sobol (Mitchell Cancer Institute, University of South Alabama, Mobile, AL) for Pol$λ^{-/-}$ MEFs. This work was supported by the National Institute of General Medical Sciences (R01 GM126087).

## Author Contributions

J-H Yoon: resources, data curation, software, formal analysis, validation, investigation, visualization, methodology, and writing—review and editing.
D Basu: formal analysis, investigation, and methodology.
K Sellamuthu: data curation, investigation, and methodology.
RE Johnson: data curation, formal analysis, investigation, and methodology.
S Prakash: conceptualization, formal analysis, supervision, validation, visualization, project administration, and writing—original draft, review, and editing.
L Prakash: resources, data curation, formal analysis, funding acquisition, validation, methodology, project administration, and writing—original draft.

## Conflict of Interest Statement

The authors declare that they have no conflict of interest.

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
