## [Reviewer comments · Life Science Alliance]

Life Science Alliance

A novel role of DNA polymerase λ in translesion synthesis in conjunction with DNA polymerase ζ

Jung-Hoon Yoon, Debashree Basu, Karthi Sellamuthu, Robert Johnson, Satya Prakash, and Louise Prakash

DOI: <https://doi.org/10.26508/lsa.202000900>

Corresponding author(s): Louise Prakash, University of Texas Medical Branch

Review Timeline:

Submission Date:	2020-08-31
Editorial Decision:	2020-11-02
Revision Received:	2020-12-11
Editorial Decision:	2020-12-23
Accepted:	2021-01-06

Scientific Editor: Shachi Bhatt

Transaction Report:

November 2, 2020

Re: Life Science Alliance manuscript #LSA-2020-00900-T

Dr. Louise Prakash
University of Texas Medical Branch
Biochemistry and Molecular Biology
301 University Blvd.
Rm 6.104 Med. Res. Bldg.
Galveston, TX USA 77555-1061

Dear Dr. Prakash,

Thank you for submitting your manuscript entitled "A novel role of DNA polymerase λ in translesion synthesis in conjunction with DNA polymerase ζ " to Life Science Alliance. The manuscript was assessed by expert reviewers, whose comments are appended to this letter.

As you can see from the reviewers' comments below, the reviewers are provisionally enthusiastic about your findings, but do point out several crucial control experiments that need to be performed, and provide constructive suggestions to clarify the interactions between polymerases lambda and zeta. We ask that you address these points and attend to all the points made about the text (in terms of citing previous work and explaining both the caveats and interpretations of your experiments). We encourage you to revise the manuscript in accordance with these requests, and re-submit it to Life Science Alliance.

Thank you for this interesting contribution to Life Science Alliance. We are looking forward to receiving your revised manuscript.

Sincerely,

Shachi Bhatt, Ph.D.
Executive Editor
Life Science Alliance
<https://www.lsjournal.org/>
Tweet @SciBhatt @LSAJournal

- A letter addressing the reviewers' comments point by point.
- An editable version of the final text (.DOC or .DOCX) is needed for copyediting (no PDFs).
- High-resolution figure, supplementary figure and video files uploaded as individual files: See our detailed guidelines for preparing your production-ready images, <https://www.life-science-alliance.org/authors>
- Summary blurb (enter in submission system): A short text summarizing in a single sentence the study (max. 200 characters including spaces). This text is used in conjunction with the titles of papers, hence should be informative and complementary to the title and running title. It should describe the context and significance of the findings for a general readership; it should be written in the present tense and refer to the work in the third person. Author names should not be mentioned.

B. MANUSCRIPT ORGANIZATION AND FORMATTING:

Reviewer #1 (Comments to the Authors (Required)):

GENERAL

The manuscript by Jung-Hoon Yoon et al. describes a novel role for PolX DNA polymerase lambda in translesion synthesis (TLS). The function of this specialized polymerase had hitherto been essentially restricted to gap-filling DNA synthesis reactions during BER and NHEJ processes, and

the authors show here that it may also have relevant functions, either structural or catalytic, in the bypass of various types of bulky DNA lesions. This new function would be carried out in complex with another specialized DNA polymerase, DNA polymerase zeta, whose participation in TLS has been previously demonstrated by the authors and other groups. The subject of the study is timely and important to have a clear vision of the different sub-pathways through which replication can bypass DNA lesions it encounters. That said, although the main conclusions of the study are robustly supported by the experimental data, the work presents some gaps that should be resolved to give it greater cohesion and strength. See specific comments below.

POINTS TO ADDRESS

Major points:

1. The introduction is excessively short. This fact is surprisingly pronounced in the case of pol lambda, which is hardly described in this section, and, paradoxically, is the main player throughout the manuscript. In this context it would also be highly recommended that the appropriate citations be included, since, for example, studies originally describing the involvement of pol lambda in BER were performed by Blanco and Kunkel laboratories, long before Braithwaite et al. confirmed these results with extracts from murine models. On the other hand, although in this work the authors accumulate much evidence of the novel activity of pol lambda in the bypass of DNA lesions in vivo, they might also take into account throughout their ms some previous work reporting pol lambda's capability as a TLS polymerase in some settings: e.g. reported pol lambda-mediated TLS at abasic sites (Maga et al. JBC 2002), or thymine glycol bypass in gapped DNA in vitro (Belousova et al. Biochemistry 2010). The fact that these studies have been carried out in vitro not only does not detract from the work presented here, nor does it affect its novelty, but supports it.

2. The main weakness of the study is found in the poor description of the molecular interactions of pol lambda and pol zeta. I firmly believe that improving this section, by supporting this interaction with some other experimental approaches, would greatly strengthen the work. In this regard, I have several comments/concerns:

- In the in vitro interaction experiment (Figure 1) the GST-alone control is missing; this is required as an undoubted confirmation of Pol lambda and Rev7 interaction. Also, perhaps the authors could include Rev3 in their analysis, or, in its absence, comment on why they have chosen this subunit and no other.

- Given that one of the strengths in the study is the structural (as a scaffold) function of pol lambda in the complex that it forms with pol zeta, I think the authors should provide more data on which part of the protein is responsible of this. Pol lambda is organized in three different structural and functional domains (N-terminal BRCT, S/P-rich linker and C-terminal catalytic β core), and it would be very interesting to know whether the formation of pol lambda/pol zeta complex requires or not the BRCT domain, given its well-known function in the establishment of protein-protein interactions. Although this domain has been shown as essential for pol lambda's role in NHEJ, this is very localized to a few amino acids, hence it would be important to know if this domain is required or not for the physical and functional interaction with pol zeta in any of TLS reactions evaluated by the authors.

- On the other hand, a demonstration that pol lambda/pol zeta interaction also takes place in vivo must be presented. This could be done by direct co-immunoprecipitation and/or

immunofluorescence using the corresponding antibodies separately (co-localization) or together (proximity ligation assays can work nicely in these cases). In this sense, it would be very interesting if this proximity (for example in the case of using PLAs) can be detected with endogenous proteins, and it would be equally important to support authors' conclusions if in vivo interaction increased in response to UV treatment.

3. To support that pol lambda/rev7 complex is assembled at replication forks stalled as a consequence of UV lesions - as claimed by the authors (page 13) - it is also necessary to show a control for this scenario in the IF assays, such as co-localization with ub-PCNA or phospho-RPA. There are even specific antibodies that recognize UV-mediated lesions (e.g. CPD and 6-4PP antibodies are available at CosmoBio) that could be used to clearly demonstrate a co-localization of this novel lambda/zeta complex with UV-induced DNA lesions.

4. Regarding IF experiments shown in Figure 3, the results would gain a lot if higher quality images are shown: the selection of captures could be improved, and DAPI images are tremendously overexposed, especially those on left panels.

5. I wonder if the authors could incorporate into their ms some mention to the fact that the plasmid system to study TLS does not allow to discriminate between "co-replicative" and "post-replicative" events. I think this is very relevant, especially considering that pol lambda is highly specialized to use gapped DNA substrates. In this sense, an attractive possibility to explain the specific requirement of pol lambda's catalytic activity opposite to 6-4PP lesions would lie in both its affinity for this type of gapped DNA (having this lesion in the template) and its particular ability to scrunch such template (García-Díaz et al NSMB 2010, <https://pubmed.ncbi.nlm.nih.gov/19701199/>), which, at least in undamaged substrates, allows it to accommodate a few uncopied template bases in an extrahelical position within a binding pocket that comprises three conserved amino acids.

The existence of the co- and post-replicative scenarios could also be discussed in relation to the role of PCNA in the TLS process involving pol lambda/pol zeta complex, since both proteins have been shown to interact with it.

6. According to the models presented by the authors in the supplementary figures (by the way, at least Figure S1 could accompany the main text) all TLS pols would have the same probability of acting on these substrates. Is this correct? If not, perhaps the authors could play with the thickness of the arrows, to identify which routes are the main and which are the alternative ones. Even they could also include a comment about it in the discussion.

Minor points:

=> Authors should be careful to overstate similarity between the plasmid system they use and the genomic context. Although it is true that many of their results are consistent, it must be recognized that in the genomic context there are features not seen in the plasmid (or viceversa), such as topology, epigenetic marks or association with proteins (including chromatin state).

=> Figure 2: if the figure represents the mutational spectra of deficient cells in either Rev3 or pol lambda, it would be preferable to use a nomenclature that indicates those deficiency conditions. As it is labeled, it seems that mutations shown are responsible for the activity of these proteins (not the result of their deficiency).

=> Perhaps the authors could hypothesize somewhere why the presence of a "scaffold" factor (other than usual PCNA) is necessary for a TLS polymerase during TLS; i.e. what does pol lambda provide that the "master scaffold" PCNA can't do.

=> At least two of the references in the list do not include their authors.

=> Figure S3b: How do the authors explain the double band that appears in the Western blot in the right panel? Such an obvious result requires an explanation, even in the figure legend.

=> The M&M section should be carefully reviewed (along with the figure legends), as there is a lack of information in some parts that I consider necessary to facilitate understanding: i.e. an adequate list of the cell lines used in the study should be provided, including the minimum specifications to work with them, specifications of images and microscopy details (the scale bars in the images are missing, microscopy magnifications at which the pictures have been taken too), source and purification degree of the oligonucleotides, etc.

=> Typo, page 4, line 4 => maybe "effects" should be "affects"

Reviewer #2 (Comments to the Authors (Required)):

In this manuscript, the authors investigate the role of Pol lambda (Pol L) in assisting translesion synthesis (TLS) on various DNA lesions, focusing more on UV-induced lesion, in cooperation with different TLS Pols. Their data suggest that PolL acts as a scaffolding protein for TT-dimer bypass and as an inserter across (6-4) TT photoproducts, in cooperation with Pol zeta (PolZ). This is a novel finding and potentially opening new lines of investigation about PolL roles in normal and cancer cells.

The authors used an assay employing a plasmid with the SV40 origin of replication containing specific lesions to measure the frequency of bypass in cells where different Pols have been silenced alone or in pairs. In addition, they measured the mutation frequency of TLS depending on the absence/presence of different Pols. Overall their data concur in showing the existence of a PolL-PolZ dependent TLS pathway distinct from Rev1, Pol eta and Pol theta pathways.

By using IF analysis, they showed that Pol L is required to mediate repair foci formation of Pol Z upon UV irradiation and viceversa.

Overall, the data are solid, but some experiments should be added to make the conclusions stronger. In addition, some controls are missing.

MAJOR POINTS

1) The novel hypothesis reported in this manuscript is that Pol L acts as a scaffolding protein for PolZ-mediated bypass of a cis-syn TT-dimer, while it acts as an inserter opposite a (6-4)TT-PP for further elongation by Pol Z. This hypothesis is well substantiated by the analysis performed, however a stronger biochemical investigation should be performed. The experiment shown in Figure 1 B indicates that PolL can be both an inserter and an elongator across a (6-4) PP-TT. In such a simple assay a similar result might be obtained with a cis-syn TT dimer? In addition, what is the effect of PolZ addition to the reaction in terms of efficiency of bypass? To directly prove the

authors' hypothesis a minimal reconstitution of a two-pols mechanism of bypass (i.e. inserter+extender) on at least (6-4) PP-TT is recommended. In addition, it would be interesting, if within reach of the authors, to see the effects of adding the catalytically inactive mutant of Pol L in an in vitro elongation reaction with PolZ and a cis-syn TT dimer containing substrate.

2) In commenting Figure 1 B, the authors state that PolL inserts dAMP opposite the 5'T of a TT dimer less efficiently than opposite the 3'T. Based on lane 8, it seems instead that the elongation from the dAMP opposite the 3'T is quite efficient leading to robust incorporation of a second dAMP opposite the 5'T. In any event, there are no experiments designed to specifically address the elongation efficiency, which would require either a time course experiment to measure the apparent rate of incorporation vs elongation or, even better, the use of a second substrate terminating with an A already annealing opposite the 3'T, to monitor the incorporation opposite the 5'T. This is not a trivial point. In the hypothesis that PolL is an extender and PolZ the elongator, a difference in insertion vs elongation efficiency between 3'T and 5'T might represent a kinetic checkpoint allowing the polL/PolZ switch, since PolZ extends from the nucleotide inserted opposite the 3'T. This could be investigated in vitro through properly designed experiments to further prove the authors' hypothesis.

3) A negative control (Pol L + GSH-beads) for the experiment shown in Figure 1 D is required to exclude unspecific binding.

4) Many important conclusions are based on the differences among the TLS% reported in Tables 1-4 and 6. However there is no indication of their statistical significance. This should be provided, along with the description of the statistical method used, for example between the control (NC or WT) and the experimentally modified (siRNA) cells.

MINOR POINTS

1) For the experiment shown in Figure 3 please specify the number of cells analysed for each condition (for example, a 3% of PolL foci containing cells value translates into how many total cells with/without foci?). Maybe a table with the raw data could be added as Supplementary Information.

2) The first paragraph of the Result contains no experimental data but is a sort of second introduction. I think it should either reduced or shifted to the introduction section, since it does not properly belong to the Results.

Dr. Shachi Bhatt
Executive Editor
Life Science Alliance

Re: Life Science Alliance manuscript # LSA-2020-00900-T

Dear Dr. Bhatt:

Thank you very much for your e-mail of November 2, 2020 regarding the review of our paper entitled "A novel role of DNA polymerase λ in translesion synthesis in conjunction with DNA polymerase ζ ." While both the Reviewers are very positive, they raised certain points for us to consider. We have revised the manuscript in response to the Reviewers' comments. In the revision, we include the control experiments, cite previous work and explain the caveats and interpretations of experiments. Our point by point response addressing the Reviewers' comments and the changes we have made in the manuscript are noted below.

Response to Reviewer #1

Major points

(1) The Reviewer wants us to cite studies on Pol λ involvement in BER by Blanco and Kunkel. In the revision, we cite their work in the second ¶ on p. 4. The Reviewer also wants us to include citations of Maga et al, JBC, 2002 and Belousova et al Biochemistry 2010 studies. We refer to these studies as well in the second ¶ on p. 4.

(2) In this comment, the Reviewer wants us to include additional data for interaction of Pol λ with Pol ζ .
(a) In response to the Reviewer's concern that in the *in vitro* interaction experiment (Figure 1), the GST-alone control is missing, in the revision we show these data in Fig. 5B. Human Rev3 is a very large protein comprised of 3130 amino acids and it has not been possible to express and purify this protein. For this reason, we are unable to include human Rev3 in this or any other biochemical analysis. The other subunits which are components of the Pol λ /Pol ζ ensemble remain to be identified.

(b) The Reviewer wants to know whether or not the formation of Pol λ /Pol ζ complex requires the BRCT domain, given its well-known function in the establishment of protein-protein interactions. In response to this comment, in Fig. 5B we show that the N-terminally deleted Pol λ -NTD which lacks the BRCT domain interacts physically with the Rev7 subunit of Pol ζ . These results are described on p. 14 in ¶ 1. In this comment, the Reviewer also wants to know if the BRCT domain is required for the functional interaction of Pol λ with Pol ζ in any of the TLS reactions evaluated by the authors. In response to this query, we analyze the effects of BRCT deleted (245-575) Pol λ on TLS opposite a (6-4) TT photoproduct, and in Table 6 we show that the lack of the BRCT domain does not affect Pol λ function in Pol ζ dependent TLS opposite this photoproduct. We discuss these results on p. 14 in the last ¶ and continuing onto p. 15. Altogether, our data show that the BRCT domain is not required for the physical and functional interaction of Pol λ with Pol ζ .

(c) In response to the next part of comment 2, in which the Reviewer wants us to provide evidence for Pol λ /Pol ζ interaction by co-immunoprecipitation and/or immunofluorescence, in Fig. 5C, we include evidence for co-immunoprecipitation of Pol λ with the Rev7 subunit of Pol ζ in chromatin fractions isolated from UV irradiated cells. We discuss these results on p. 14 in ¶ 2..

(3) To support that the Pol λ /Rev7 complex is assembled at replication forks stalled as a consequence of UV lesions, the Reviewer wants us to show evidence for co-localization of Pol λ /Rev7 with ub-

PCNA. In response to this comment, we provide evidence for co-immunoprecipitation of Pol λ and Rev7 with ub-PCNA in UV irradiated cells in Fig. 5C and discuss these results on p. 14 in ¶ 2.

(4) Regarding IF experiments shown in Fig. 3, in the revision in Fig. 4, we have attended to all the concerns of the Reviewer.

(5) Regarding this Reviewer's comment on whether Pol λ 's catalytic activity opposite the (6-4) PP lies in its affinity for gapped DNA, our evidence has indicated that TLS in normal human cells operates in conjunction with the replication fork and not post-replicatively in gaps. Thus, TLS by Pol λ or by any other TLS Pol would occur co-replicationally in normal cells. We briefly allude to this evidence in the first ¶ on p. 5.

The physical interaction of TLS Pols with PCNA, including Pol λ , would be necessary for their ability to carry out lesion bypass regardless of whether TLS occurs co-replicationally or post-replicationally.

(6) In response to this comment, in Fig. S1, which we now include in the main text as Fig. 1 as suggested by this Reviewer, we have changed the thickness of the arrows to correspond with their respective roles. We have also moved previous Fig. S2 to the main text as Fig. 7. Since the respective roles of TLS Pols are discussed in the Results section, any further comments in the Discussion would be redundant and rather out of place.

Minor points

(1) In response to this comment, we now qualify the similarity between the plasmid system and the genomic context. At the end of ¶1 on p. 5, we include the statement "Although not all the complexities in the genomic context such as topology, chromatin state, or epigenetic modifications will be reflected in the plasmid system, the basic TLS mechanisms remain the same in the genomic context as in the plasmid system (Yoon et al., 2019b).

(2) Fig. 2: In Fig. 3 in the revised version, we identify mutational spectra for Rev3 or Pol λ deficient cells by denoting it by Rev3 siRNA or Pol λ siRNA.

(3) Ub-PCNA vs. Pol λ would affect TLS in very different ways. Whereas ub-PCNA is a prerequisite for the stalling of replication forks (RFs) at DNA lesion sites; and very likely, for the ejection of the replicative Pol from PCNA and for the subsequent placement of TLS Pols at the stalled RFs, Pol λ would affect the formation of multiprotein ensemble of Pol ζ . On p. 14 at the end of ¶ 2, we comment on the role of ub-PCNA. On p. 22 in ¶ 1, we elaborate on how the Pol λ /Pol ζ multiprotein ensemble could affect the fidelity and efficiency of both these Pols for TLS opposite a diverse array of DNA lesions.

(4) We now include the authors in the two references.

(5) Fig. S3b: In the legend for Fig. S1B in the revision, we include a reference for the phosphorylation of Pol λ and suggest that the upper band is likely phosphorylated Pol λ .

(6) On p. 25 in Materials and Methods, we include a list of the cell lines used including the specification to work with them. In Fig. 4, we provide the scale bar for the images and in the Fig. 4 legend, we include the magnification used for the images. Oligonucleotides were purchased from various companies. In the section on DNA polymerase assays on p. 24, we have added a reference pertaining to the source and construction of oligos containing a *cis-syn* TT dimer or a (6-4) TT photoproduct and have added a statement that all oligonucleotides were PAGE purified.

(7) Typo, p. 4, line 4. We use “effects” and not “affects” because we imply an active role for Rev1 in the formation of multiprotein ensemble of Y-family Pols.

Reviewer #2

Major points

(1) In this comment, the Reviewer state “The experiment shown in Figure 1 B indicates that PolL can be both an inserter and an elongator across a (6-4) PP-TT. In such a simple assay a similar result might be obtained with a *cis-syn* TT dimer?” In response to this comment, we now include Fig. S2 which shows that Polλ is completely inactive in TLS opposite a *cis-syn* TT dimer.

In the second part of this comment, the Reviewer wonders whether adding Polζ to the reaction would affect the efficiency of bypass and suggests reconstituting the two-Pols mechanism of bypass for the (6-4) TT lesion. The reconstitution of such a two-Pol mechanism is not a trivial task as that would require the purification of the entire multiprotein ensemble of Polλ with Polζ. Moreover, since for TLS in normal human cells, Polζ requires Polλ and not Rev1, we expect the composition of Polλ/Polζ to differ from that of Rev1/Polζ. Hence, that would require the identification of those novel Polζ subunits (We have some preliminary evidence for that being the case). Additionally human Rev3 (catalytic subunit of Polζ) is a very large protein (344 kDa) containing 3130 amino acids and it has not been possible to purify even the human Rev3/Rev7 complex. However, to allay the concern of this Reviewer that Polζ polymerase activity may not be required, we now include Table 8 in which we provide evidence that Polζ polymerase activity is in fact required for TLS through the (6-4) TT PP in human cells. Furthermore, the fact that purified human Polλ inserts nts opposite (6-4) TT PP and that purified yeast Polζ lacks the ability to insert nts opposite the 3'T of the photoproduct strongly suggests that following nt insertion opposite both the pyrimidines of the photoproducts, Polζ would extend synthesis. Whereas the broad outlines of TLS Pols function in the insertion vs. extension steps could be construed from studies done with individual Pols, the elucidation of details of their mechanism of action would require the reconstitution of functional multiprotein ensembles of TLS Pols – a very long and arduous task.

In the last section of this comment, the Reviewer suggests that it would be interesting, if within reach, to see the effects of adding the catalytically inactive mutant of Polλ in an *in vitro* elongation reaction with Polζ on a *cis-syn* TT dimer containing substrate. However, as shown in Fig. 1A, since Polλ/Polζ dependent TLS opposite a *cis-syn* TT dimer would require Polθ at the insertion step, the reconstitution of such a system would require the purification of Polθ multiprotein ensemble as well as Polλ/Polζ ensemble – clearly not possible now. Nevertheless, our genetic evidence in Table 3 shows that Polλ DNA polymerase activity is not required for TLS opposite a *cis-syn* TT dimer, and thus, only its scaffolding role is required.

(2) In the first part of this comment, the Reviewer states that “Based on lane 8, it seems instead that the elongation from the dAMP opposite the 3'T is quite efficient leading to robust incorporation of a second dAMP opposite the 5'T.” The data in lane 8 of Fig. 2B in the revision shows that overall, Polλ inserts dATP opposite both the 3' and 5'T of the photoproduct less efficiently than opposite the undamaged Ts, as in indicated from the use of almost all the DNA substrate in lane 2. In the second part of this comment, the Reviewer wants us to investigate the kinetics of insertion and extension reactions opposite the 3'T and 5'T of the photoproduct of Polλ vs. Polζ. As we say in our response to comment 1 of this Reviewer, it has not been possible to purify even the human Rev3 protein and the proper understanding of Polλ/Polζ roles in TLS would require the reconstitution of Polλ/Polζ multiprotein ensemble. That is clearly out of scope for this study.

(3) In the revision, in Fig. 5B we provide the data for the negative control of Pol λ with GST beads.

(4) For all the TLS data, we now provide statistical analyses of the data in Supplemental Tables S1-S6.

MINOR POINTS

(1) In the Fig. 4 legend in the revision, we state that for each analysis ~500 cells were analyzed from 3 independent experiments. Therefore, a 3% of Pol λ foci translates to ~15 cells with foci and ~485 cells without foci.

(2) We think it is important to point out the relevance of TLS analyses with the plasmid system for TLS in the genomic context, and the first paragraph of the Results section is the most appropriate place to do so. In fact, in the first minor Point, Reviewer 1 wants us to elaborate on the features of TLS in the plasmid system vs. the genomic context.

We thank the Reviewers for their help in improving the manuscript.

We hope that you will find the additions we have made in the revision and our response to the Reviewers' comments satisfactory.

Sincerely,

Louise Prakash

December 23, 2020

RE: Life Science Alliance Manuscript #LSA-2020-00900-TR

Dr. Louise Prakash
University of Texas Medical Branch
Biochemistry and Molecular Biology
301 University Blvd.
Rm 6.104 Med. Res. Bldg.
Galveston, TX USA 77555-1061

Dear Dr. Prakash,

Thank you for submitting your revised manuscript entitled "A novel role of DNA polymerase λ in translesion synthesis in conjunction with DNA polymerase ζ ". We would be happy to publish your paper in Life Science Alliance pending final revisions necessary to meet our formatting guidelines.

To upload the final version of your manuscript, please log in to your account: <https://lsa.msubmit.net/cgi-bin/main.plex>

A. FINAL FILES:

B. MANUSCRIPT ORGANIZATION AND FORMATTING:

Sincerely,

Shachi Bhatt, Ph.D.
Executive Editor
Life Science Alliance
<https://www.lsjournal.org/>
Tweet @SciBhatt @LSAjournal

Reviewer #1 (Comments to the Authors (Required)):

Dear editor,

After careful reading the revised version of the manuscript by Dr. Prakash and colleagues entitled "A Novel Role of DNA Polymerase λ in Translesion Synthesis Along with DNA Polymerase ζ " for Life Science Alliance (LSA), I can certify that the authors have adequately addressed most of the concerns that arose after my first review of their manuscript.

This piece of work describes a novel and remarkable role for PolX DNA polymerase lambda in translesion synthesis. The function of this specialized polymerase had hitherto been essentially restricted to gap-filling DNA synthesis reactions during BER and NHEJ processes. The authors show in this study that Pol λ can also display an interesting role as an integral scaffolding component of Pol ζ complex to promote, in a predominantly error-free manner, the bypass of some bulky DNA lesions in human cells. The study has been developed in a conscientious way, as in general those performed in the laboratory of the authors, main experts in the field. Their findings and conclusions are important to the field, hence, the study achieves high significance and conclusiveness. The main concerns after my first review have been carefully addressed in the new version, and therefore, the article has improved significantly. For all this, in my opinion the manuscript now deserves its publication in Life Science Alliance.

Reviewer #2 (Comments to the Authors (Required)):

In this study, the authors identify a novel role of Pol lambda in translesion synthesis (TLS), as a scaffolding protein for Pol zeta. The authors demonstrated such a role with genetic and biochemical assays over a number of different DNA lesion. In response to my previous requests, the authors provided novel data and further solid rationale for their approach. Their data open new perspectives on how to further dissect the complex network of interactions among the different actors in the TLS pathways and provide evidence for the fact that specialized Pols might also have additional roles besides their catalytic activity, hence acting as auxiliary factors. I have no further requests.

January 6, 2021

RE: Life Science Alliance Manuscript #LSA-2020-00900-TR

Dr. Louise Prakash
University of Texas Medical Branch
Biochemistry and Molecular Biology
301 University Blvd.
Rm 6.104 Med. Res. Bldg.
Galveston, TX USA 77555-1061

Dear Dr. Prakash,

Thank you for submitting your Research Article entitled "A novel role of DNA polymerase λ in translesion synthesis in conjunction with DNA polymerase ζ ". It is a pleasure to let you know that your manuscript is now accepted for publication in Life Science Alliance. Congratulations on this interesting work.

DISTRIBUTION OF MATERIALS:

Again, congratulations on a very nice paper. I hope you found the review process to be constructive and are pleased with how the manuscript was handled editorially. We look forward to future exciting submissions from your lab.

Sincerely,

Shachi Bhatt, Ph.D.

Executive Editor

Life Science Alliance

<https://www.lsjournal.org/>
